# Synthetic Data Generation with Large Language Models for Text Classification: Potential and Limitations

**Zhuoyan Li[1], Hangxiao Zhu[2], Zhuoran Lu[1], Ming Yin[1]**
[1]Purdue University
[2]Washington University in St. Louis
{li4178, lu800, mingyin}@purdue.edu
hangxiao@wustl.edu

## Abstract

The collection and curation of high-quality training data is crucial for developing text classification models with superior performance, but it is often associated with significant costs and time investment. Researchers have recently explored using large language models (LLMs) to generate synthetic datasets as an alternative approach. However, the effectiveness of the LLM-generated synthetic data in supporting model training is inconsistent across different classification tasks. To better understand factors that moderate the effectiveness of the LLM-generated synthetic data, in this study, we look into how the performance of models trained on these synthetic data may vary with the *subjectivity* of classification. Our results indicate that subjectivity, at both the task level and instance level, is negatively associated with the performance of the model trained on synthetic data. We conclude by discussing the implications of our work on the potential and limitations of leveraging LLM for synthetic data generation[1].

## 1 Introduction

Today, machine-learning-powered text classification models have been widely applied in diverse applications such as detecting biased or toxic language on online platforms (Wiegand et al., 2019) and filtering spam emails (Jindal and Liu, 2007). However, the performance of these models largely depends on the quality of the training data. This poses a substantial challenge in practice, especially when models need to be built for a novel task domain or to incorporate new classification categories, as the training data collection and curation process is often costly, time-consuming, and complex.

Meanwhile, with the recent advancements in large language models (LLMs), researchers have started to explore the potential of utilizing LLMs for generating synthetic data tailored to specific

tasks and augmenting the training data in low-resourced data settings (Kumar et al., 2020; Yoo et al., 2021; Hartvigsen et al., 2022; Sahu et al., 2022). Most recently, a few studies also investigate into the feasibility of generating a synthetic dataset from scratch using LLMs to support zero-shot learning (Ye et al., 2022; Wang et al., 2021; Tang et al., 2023; Gao et al., 2023). While LLM-based data augmentation is often found to outperform other data augmentation methods in boosting the model performance, mixed results are reported regarding whether the LLM-generated synthetic data can effectively support model training to enable a level of model performance that is comparable to models trained on the data collected in the real world and carefully annotated. This leaves uncertainty for researchers and practitioners in deciding whether to rely on LLMs for synthetic data generation or to proceed with the traditional data collection and curation pipeline when they need to construct a text classification model for a new task. Naturally, one may wonder *what factors might moderate the effectiveness of LLM-generated synthetic data in facilitating successful model training*.

We conjecture that one such factor could be the *subjectivity* of classification tasks. Indeed, language is inherently subjective and interpretive (Benveniste, 1971; Wiebe et al., 2004). Previous research has showed that people often perceive the same text in different ways because of their personal biases and perspectives (Sap et al., 2021; Li et al., 2022; Gordon et al., 2022). Thus, achieving high model performance for classification tasks with high subjectivity seems to impose a greater demand on the training data in reflecting the richness and nuances present in human language, and the extent to which LLM-generated synthetic data can acomplish this objective is unclear.

Thus, in this paper, we formally evaluate the effectiveness of LLM (i.e., the cutting-edge GPT-3.5-Turbo model) in generating synthetic data to

---

[1]The collected human annotations are available at huggingface.co/datasets/xfleezy/human_annotation_emnlp23.

support model training for different text classification tasks. We adopt two approaches for synthetic data generation—a *zero-shot* setting in which the LLM is directly prompted to generate text instances with different labels of interests, and a *few-shot* setting in which a few real-world data instances are provided as examples to guide the LLM in generating the synthetic data. We conduct two evaluation studies, each corresponding to one dimension of subjectivity—the first study examines the effectiveness of the synthetic data on 10 types of classification tasks and explores how it varies with the *task-level subjectivity* (i.e., whether this type of classification task is subjective); the second study concerns that given a specific classification task, how the performance of a model trained on synthetic data changes with the *instance-level subjectivity* (i.e., whether people tend to disagree with each other on the label of this task instance). Our findings suggest that across the 10 types of classification tasks that we have considered in this study, models trained on the LLM-generated synthetic data generally perform worse than those trained on the real-world data, yet guiding LLM's synthetic data generation process with a small amount of real-world data (i.e., as done in the few-shot data generation setting) can improve the effectiveness of the data generated. Moreover, we find that the performance of models trained on the LLM-generated synthetic data is very close to those trained on the real-world data for tasks with low subjectivity (e.g., news topic classification, spam email detection), while the performance decrease is much bigger on tasks with high subjectivity (e.g., humor or sarcasm detection). Finally, even within the same type of classification task, models trained on the LLM-generated synthetic data tend to exhibit a higher level of performance on those task instances with lower subjectivity, for which human annotators exhibit a higher level of agreement in their annotation.

Together, our study provides important experimental evidence regarding the potential and limitations of using LLMs to generate synthetic data for text classification tasks. We conclude by discussing the implications, limitations, and future work of our study.

## 2 Related Work

**Generative AI in synthetic data generation.** Recent advancements in generative AI have motivated numerous studies to explore the potential of leveraging generative models to create synthetic data for training machine learning models, especially for computer vision (CV) and natural language processing (NLP) tasks. In the realm of CV, several works have utilized GAN-based models (Karras et al., 2019) or diffusion models (Nichol et al., 2021) to generate synthetic data for image recognition (Besnier et al., 2020; He et al., 2022) or object segmentation (Zhang et al., 2021). Similarly, in the NLP field, researchers have also probed into the capacity of language models in generating synthetic data for various text classification tasks (Kumar et al., 2020; Chung et al., 2023; Sahu et al., 2022; Yoo et al., 2021; Ye et al., 2022; Wang et al., 2021), with mixed results reported regarding the effectiveness of the synthetic data generated. In this study, we aim to obtain a better understanding of *when* the synthetic data generated by language models can lead to effective model training, and we focus on exploring the role of task subjectivity in moderating the effectiveness of the synthetic data.

**Large language models.** Based on the Transformer architecture (Vaswani et al., 2017), large language models (LLMs) have facilitated remarkable progress in the field of natural language processing. The utilization of bidirectional contexts in the BERT model (Devlin et al., 2018) has resulted in superior performance across a wide range of tasks. Building on this, OpenAI's GPT series, comprising of models like GPT-2 (Radford et al., 2019), the colossal GPT-3 (Brown et al., 2020) with an impressive 175 billion parameters and the most recent GPT-4 (OpenAI, 2023), pushed the boundaries of possibilities of LLMs. These models exhibit remarkable proficiency in generating high-quality human-like text (Clark et al., 2021; Dou et al., 2021; Zhou et al., 2023), showcasing capabilities in rudimentary reasoning (Wei et al., 2021), translation (Brown et al., 2020), scientific synthetic data generation (Hämäläinen et al., 2023), and code generation (Mcnutt et al., 2023). In this study, we focus on leveraging the cutting-edge GPT-3.5-Turbo model[2] to explore its capabilities and limitations in synthesizing data for text classification tasks with different subjectivity levels.

---

[2]We used GPT-3.5-Turbo as the foundational model to generate synthetic data because at the time of this study, an official API for the more advanced GPT-4 model was not yet available from OpenAI.

## 3 Methodolgy

In this section, we outline the procedure we have followed when leveraging the large language model to generate the synthetic training data for text classification. We consider two data generation settings in this study, i.e., the *zero-shot* setting and the *few-shot* setting.

### 3.1 Zero-shot Synthetic Data Generation

Under the *zero-shot* synthetic data generation setting, given a text classification task, we assume that the real-world data in the form of "text-label pairs" do not exist. Thus, in order to obtain synthetic training data for the text classification task, two sequential prompts are constructed and supplied to the pretrained large language model (i.e., the GPT-3.5-Turbo model). First, a customized "context prompt" relevant to the targeted domain of interest is used to set the context. For example, in the case of the IMDB movie review classification task (Maas et al., 2011), the customized context prompt used is "Imagine you are a movie reviewer on the IMDB platform". This prompt aims to encourage the LLM to generate synthetic data that resemble the real texts produced in the targeted domain. After the context is set, a second prompt, i.e., the "data generation prompt", is provided to the LLM, instructing the model to generate texts with a specific style, label (with respect to the classification task of interest), and word limit. For example, for the IMDB movie review classification task, the style of the text is a movie review, and the label is a targeted sentiment conveyed by the review (i.e., "positive" or "negative"). To further enhance the diversity of the generated data, after the generation of every $n$ data points (i.e., texts of targeted styles along with their labels)[3], we provide a "diversity prompt" to the LLM—"Can you provide something more diverse compared to the previously generated data?"—aiming to increase the diversity of the synthetic data generated.

### 3.2 Few-shot Synthetic Data Generation

Under the *few-shot* synthetic data generation setting, we assume that a small amount of real-world data are available for the text classification task. These data points can then serve as the examples

---

[3]To increase data diversity while maintaining a reasonable data generation speed, $n$ is set to 10 for generating short texts (i.e., texts with a maximum length of 30 words), and 1 for generating longer paragraphs.

for the large language model in the data generation process, which can potentially provide LLM with insights of the patterns exhibited in the real-world data. We again start the data generation process by using a context prompt to set the context. However, different from that in the zero-shot setting, here, each time before we instruct the LLM to generate a piece of text, we first provide the model with a few randomly sampled real-world data instances (including both the text and the label) as the examples. To keep the LLM from merely rephrasing the provided examples, an additional prompt is used to impose a constraint on the LLM in generating the synthetic data (i.e., "You should imitate the example I have provided, but you cannot simply modify or rewrite the example I have given.").

For more details about prompts used for generating data for each type of text classification task, please refer to the App. D.

## 4 Evaluation I: Comparison Across Different Types of Tasks

In our first evaluation study, we investigate into how well the synthetic data generated by LLM under both zero-shot and few-shot settings can support effective model training for different types of text classification tasks. We are especially interested in comparing the model performance between those trained on the real-world data and on the LLM-generated synthetic data, and in understanding how the performance of those models trained on the LLM-generated synthetic data varies with the subjectivity of the text classification task.

### 4.1 Datasets and Tasks

We experiment with 10 representative datasets covering a variety of text classification tasks: AG's news (Zhang et al., 2015b), IMDB reviews (Maas et al., 2011), SMS spam (Almeida et al., 2011), Financial phrase bank (Malo et al., 2014), Reddit emotion (Demszky et al., 2020), Relation classification (Gao et al., 2019), Tweet irony speech (Van Hee et al., 2018), Tweet emotions (Mohammad et al., 2018), Sarcasm news (Misra and Arora, 2023, Misra and Grover, 2021), and Humor speech (Annamoradnejad and Zoghi, 2020). See App. A.1 for detailed descriptions of datasets and the corresponding text classification tasks. These datasets are selected with the goal of spanning a wide range of task subjectivity in mind. For example, we conjecture that classifying the news topic

category (e.g., as that in the AG's news dataset) is relatively objective, while determining whether texts are humorous (e.g., as that in the Humor speech dataset) is quite subjective (Veatch, 1998).

## 4.2 Task-level Subjectivity Determination

To formally determine the subjectivity levels of different text classification tasks, we first conduct a crowdsourced study to collect subjectivity judgements from the crowd.

**Study procedure.** We adopt a comparative approach to collect crowdsourced subjectivity judgements in this study. Specifically, we recruited crowd workers from Amazon Mechanical Turk (MTurk), and each worker was asked to complete a sequence of 10 subjectivity judgement tasks. In each task, we randomly sampled a pair of text classification tasks from the 10 tasks that we considered in this evaluation, and we presented to the worker the task description, label description, and task examples for each task in the pair. Then, the worker was asked to determine which text classification task in the pair was more objective, with "objectivity" of a task defined as "the classification of a piece of text is based on clear, identifiable features in the text (e.g., keywords or phrases), and can be done without being affected by any personal interpretation of the text resulted from personal biases, emotions or beliefs." The study was restricted to U.S. workers. Each worker was allowed to participate only once and received a $1.2 payment. An attention check question was included in the study to validate the worker's engagement, and only the data from workers who successfully passed the attention check were considered valid.

**Ranking task subjectivity.** After excluding responses from inattentive workers, a total of 540 pairwise subjectivity comparisons for the 10 tasks were obtained from 54 workers. For each pair of tasks, we aggregated relative subjectivity judgments made on this pair to determine which task was perceived as more subjective (i.e., less objective). To produce a ranking of the subjectivity of the 10 tasks, we constructed a directed graph based on the pairwise subjectivity comparisons—each task was a node in this graph, and directed edges were added between each pair of tasks, pointing from the one that was deemed as more subjective (on the aggregate level) to the one deemed as less subjective. The topological sort algorithm (Cormen et al., 2022) was then applied to this directed graph

to obtain a linear ordering of the nodes. If a cycle was detected within the graph, the corresponding tasks were considered to have the same level of subjectivity and were merged into a single meta-node before re-runing the algorithm. Our final task subjectivity ranking results are shown in Table 1.

## 4.3 Model Training

Given a text classification task, following the procedures outlined in Sections 3.1 and 3.2, 3,000 synthetic data points were generated for each candidate label under both zero-shot and few-shot settings. We then trained classification models using the real-world training data provided by the original dataset, the synthetic data generated under the zero-shot settings, and the synthetic data generated under the few-shot settings[4], respectively. Specifically, we utilized the pre-trained BERT (Devlin et al., 2018) and RoBERTa (Liu et al., 2019) models from Huggingface's transformers library (Wolf et al., 2020) as the encoders, and used the representation embeddings from the last layer of these models as the input to our classification models. The classification model itself comprised a hidden layer of 768 units and an output layer, and it was fine-tuned with a learning rate of $5e-5$ and a batch size of 64. For datasets that provided official partitions for training and test sets, we directly evaluated the classification model's performance on the test sets. Otherwise, we randomly divided the dataset into training (70%), validation (5%), and test (25%) sets[5]. Models' performance was evaluated via Macro-F1 and Accuracy scores, and they were computed by comparing the model's predictions with the gold labels provided in the test sets. To ensure the robustness of our results, all experiments were repeated three times, and the average performance across these repetitions was reported.

## 4.4 Evaluation Results

Table 1 summarizes the comparative performance of classification models trained with different data. Below, we highlight a few key observations we get from this comparison.

---

[4]Under the few-shot setting, we randomly sampled 10% of the data points from the real-world training data provided in the original dataset as the example pool to guide the LLM's synthetic data generation process, but only the sythetic data generated were used to train the models.

[5]To ensure a fair comparison, we maintained an equal size for both the real-world and synthetic training data by downsampling the dataset with a larger size.

| Dataset | Subjectivity | BERT | | | | | | RoBERTa | | | | | |
|---|---|---|---|---|---|---|---|---|---|---|---|---|---|
| | | Real-world data | | Zero-shot setting | | Few-shot setting | | Real-world data | | Zero-shot setting | | Few-shot setting | |
| | | Macro-F1 | Accuracy Score | Macro-F1 | Accuracy Score | Macro-F1 | Accuracy Score | Macro-F1 | Accuracy Score | Macro-F1 | Accuracy Score | Macro-F1 | Accuracy Score |
| AG | ⋆ | 95.3% | 95.3% | 89.3% (-6.0%) | 89.3% (-6.0%) | 91.5% (-3.8%) | 91.6% (-3.7%) | 94.6% | 94.6% | 88.6% (-6.0%) | 88.6% (-6.0%) | 92.9% (-1.7%) | 92.9% (-1.7%) |
| Relation | ⋆⋆ | 98.6% | 98.6% | 92.4% (-6.2%) | 92.7% (-5.9%) | 96.4% (-2.2%) | 96.4% (-2.2%) | 97.0% | 96.9% | 91.4% (-5.6%) | 91.6% (-5.3%) | 94.1% (-2.9%) | 94.1% (-2.8%) |
| IMDB | ⋆⋆⋆ | 87.6% | 87.6% | 81.2% (-6.4%) | 81.5% (-6.1%) | 81.1% (-6.5%) | 81.2% (-6.4%) | 89.0% | 89.0% | 81.2% (-7.8%) | 81.3% (-7.7%) | 82.4% (-1.6%) | 82.4% (-1.6%) |
| SMS spam | ⋆⋆⋆⋆ | 97.2% | 98.8% | 93.8% (-3.4%) | 95.1% (-3.7%) | 94.3% (-2.9%) | 94.8% (-4.0%) | 97.3% | 98.8% | 93.5% (-3.8%) | 95.9% (-2.9%) | 94.0% (-3.3%) | 95.7% (-3.1%) |
| Reddit emotion | ⋆⋆⋆⋆⋆ | 93.7% | 94.6% | 72.7% (-21.0%) | 74.4% (-20.2%) | 81.9% (-11.8%) | 82.0% (-12.6%) | 91.3% | 92.1% | 77.9% (-13.4%) | 78.1% (-14.0%) | 87.5% (-3.8%) | 87.7% (-4.4%) |
| Tweet irony | ⋆⋆⋆⋆⋆ | 72.2% | 73.9% | 63.4% (-8.8%) | 63.6% (-10.3%) | 81.5% (+9.3%) | 81.9% (+8.0%) | 74.0% | 75.5% | 57.8% (-16.2%) | 59.1% (-16.4%) | 83.3% (+9.3%) | 83.7% (+8.2%) |
| Tweet emotions | ⋆⋆⋆⋆⋆ | 77.7% | 81.1% | 58.1% (-19.6%) | 64.5% (-16.6%) | 64.6% (-13.1%) | 69.1% (-12.0%) | 75.8% | 78.9% | 64.6% (-11.2%) | 71.5% (-7.4%) | 66.3% (-9.5%) | 72.7% (-6.2%) |
| Sarcasm | ⋆⋆⋆⋆⋆ | 89.9% | 90.3% | 51.1% (-38.8%) | 51.2% (-39.1%) | 63.6% (-26.3%) | 64.8% (-25.5%) | 91.8% | 92.0% | 54.3% (-37.5%) | 54.3% (-37.7%) | 61.5% (-30.3%) | 63.6% (-28.4%) |
| Financial | ⋆⋆⋆⋆⋆ | 83.2% | 84.6% | 48.2% (-35.0%) | 60.7% (-23.9%) | 70.6% (-12.6%) | 74.2% (-10.4%) | 85.0% | 86.6% | 58.5% (-26.5%) | 70.3% (-16.3%) | 75.0% (-10.0%) | 78.9% (-7.7%) |
| Humor speech | ⋆⋆⋆⋆⋆ | 97.0% | 97.0% | 56.0% (-41.0%) | 61.7% (-35.3%) | 86.9% (-10.1%) | 87.0% (-10.0%) | 96.7% | 96.7% | 54.9% (-41.8%) | 60.9% (-35.8%) | 84.0% (-12.7%) | 84.0% (-12.7%) |

Table 1: Comparing the performance of classification models trained on the LLM-generated synthetic data under the zero-shot or few-shot settings, with those trained with the original real-world data, in terms of Macro-F1 (%) and Accuracy Score (%). In the "Subjectivity" column, more "⋆" symbols indicate a higher level of task subjectivity.

**Models trained on the real-world data consistently outperform those trained on the synthetic data.** Our results indicate that models trained on the original real-world data consistently outperform their counterparts trained on the synthetic data generated under either zero-shot or few-shot settings, almost for every task. In particular, with the RoBERTa model, we observe that the average improvements of the model trained on the real-world data over the models trained on zero-shot synthetic data and few-shot synthetic data are 16.9% and 6.7% in terms of Macro-F1, and 14.9% and 6.1% in terms of accuracy. Similar trends are observed with the BERT model as well.

**Guiding LLM with real-world data examples can boost the effectiveness of the synthetic data.** We also observe that models trained on those synthetic data generated under the few-shot settings almost always outperform those trained on the synthetic data generated under the zero-shot settings. For instance, for the BERT model, we see an average increase of 10.6% and 8.8% in Macro-F1 and accuracy scores, respectively, across the 10 tasks in the few-shot setting, as compared to the zero-shot setting. Similarly, with the RoBERTa model, there is an average increase of 10.3% in Macro-F1 and 8.9% in accuracy scores across the 10 tasks when the real-world data are used as examples for LLM to mimic in the synthetic data generation process. For more analysis of the few-shot synthetic data, please see App. B.2 and B.3.

**Synthetic data support more effective model training for tasks that are less subjective.** Finally, we notice that for classification tasks with relatively low levels of subjectivity (e.g., those in the AG's news, Relation classification, IMDB reviews, and SMS spam datasets), the performance difference between models trained on the synthetic data and those trained on the real-world data is remarkably small. However, for tasks with high subjectivity, the performance decrease resulted from the usage of the synthetic data is more significant—for instance, across the cluster of 6 tasks with the highest level of subjectivity in our evaluation, there is an average decrease of 27.4% and 24.2% in Macro-F1 and accuracy, respectively, comparing the BERT models trained on the zero-shot synthetic data with those trained on the real-world data. In other words, for text classification tasks that are highly objective, there is great potential in training high-performing models simply based on synthetic data generated by LLMs, but the same method falls short in generating synthetic data that can effectively support model training for highly subjective classifications. For more robustness check of this finding (e.g., on more datasets, when using different LLMs for data generation, when using alternative data generation pipelines), see App. B.4–B.10 for details.

### 4.5 Exploratory Analysis: Data Diversity

To explore the potential reasons underlying the model performance difference, we conducted an exploratory analysis on the diversity of the training data. Following Rhys Cox et al. (2021), we used the *Remote Clique Score* (i.e., the average mean distance of a data instance to other instances) and the *Chamfer Distance Score* (i.e., the average minimum distance of a data instance to other instances) to quantify the diversity of a set of data. For both metrics, higher values indicate greater data diversity. As shown in Figure 1, we find that in general, the real-world data appear to be more diverse than the synthetic data generated under the few-shot settings, which in turn seem to be more diverse than the zero-shot synthetic data. This might partially explain why models trained on the real-world data and the few-shot synthetic data tend to outperform those trained on the zero-shot synthetic data.

In addition, we also notice that compared to that on the low subjectivity tasks (i.e., AG, Relation,

| Dataset | AG | Relation | IMDB | SMS Spam | Reddit Emotion | Humor Speech | Tweet Irony | Sarcasm | Tweet Emotions | Finanical |
|---|---|---|---|---|---|---|---|---|---|---|
| Average Agreement $\overline{a}$ | 0.80 (4.2) | 0.78 (4.5) | 0.76 (7.3) | 0.73 (8.5) | 0.69 (6.6) | 0.68 (7.1) | 0.68 (6.7) | 0.64 (7.7) | 0.64 (4.6) | 0.57 (7.6) |
| Krippendorff's $\alpha$ | 0.51 | 0.43 | 0.19 | 0.27 | 0.30 | 0.06 | 0.03 | 0.01 | 0.17 | -0.03 |
| Subjectivity Level | ★ | ★★ | ★★★ | ★★★★ | ★★★★★ | ★★★★★ | ★★★★★ | ★★★★★ | ★★★★★ | ★★★★★ |

Table 2: The average instance-level annotation agreement for different types of tasks, alongside the corresponding task-level subjectivity. Numbers in parentheses in the first row represent the average number of annotations received per task instance. Higher values for both the average agreement $\overline{a}$ and Krippendorff's $\alpha$ indicate a higher degree inter-annotator agreement.

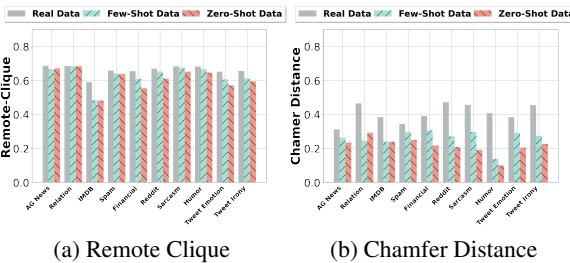

(a) Remote Clique      (b) Chamfer Distance

Figure 1: Comparing the diversity of the real-world data and the synthetic data.

IMDB, Spam), the differences in data diversity between the real-world data and the synthetic data seem to be more salient on the high subjectivity tasks (i.e., the other 6 tasks), especially in terms of the Chamfer Distance Score. In fact, a t-test shows that the decrease of the Chamfer Distance Score in the zero-shot synthetic data compared to the real data is significantly larger for the high subjectivity tasks than for the low subjectivity tasks ($p < 0.01$). This suggests that for tasks with high subjectivity, such as interpreting humor or sarcasm in language, LLMs may not be able to generate data instances that can cover the full spectrum of real-life scenarios, which may limit the performance of models trained on the synthetic data.

## 5 Evaluation II: Comparison Across Different Task Instances

In the previous section, we have discovered that the subjectivity of a task can adversely affect the performance of classification models trained on the LLM-generated synthetic data. However, even for the same type of task, the classification for each individual task instance may exhibit different levels of subjectivity as well. Naturally, one may wonder whether models trained on the LLM-generated synthetic data may show different performance on task instances of different subjectivity. We aim to explore the answers to this question in this section.

### 5.1 Instance-level Subjectivity Determination

Given a text classification task and a specific text instance, we consider the degree of *agreement among annotators* on the label of this text as a proxy for the subjectivity of this instance—a lower level of agreement means that annotators hold more divergent views, hence the task may have a higher level of subjectivity. Thus, to formally quantify the subjectivity of different instances for different tasks, we again conduct a crowdsourced study to collect instance-level annotations.

**Study procedure.** We again considered the 10 types of text classification tasks as that in the first evaluation study. For each type of task, we randomly sampled 50 text instances per category from the test set to compose our "evaluation dataset" for that task. We then recruited U.S. workers from MTurk to complete annotation tasks for those instances in our evaluation dataset. Specifically, each worker was randomly assigned to one type of text classification tasks. After going through a brief instruction of the assigned task, the worker was asked to complete 20 classification tasks of the assigned type to get a payment of $1.2, where the texts presented in these 20 tasks were randomly sampled from the evaluation dataset for the assigned type of task. Again, we included two attention check questions in our study to filter out inattentive workers. We ensured that each task instance received at least three annotations from unique MTurk workers.

**Computing instance subjectivity.** Based on annotations we obtained from attentive workers, we quantify the subjectivity level of each task instance using the fraction of annotators who agree with the majority label for the task instance, that is:

$$a_i = \frac{\max_{y \in \mathcal{Y}} \sum_{k=1}^{K_i} \mathbb{1}(r_i^k = y)}{K_i} \qquad (1)$$

where $\mathcal{Y} = \{1, \cdots, Y\}$ is the set of all possible labels, $K_i$ is the total number of annotators who labeled instance $i$, and $r_i^k$ is the $k$-th annotator's annotation on instance $i$. Intuitively, a lower value of $a_i$

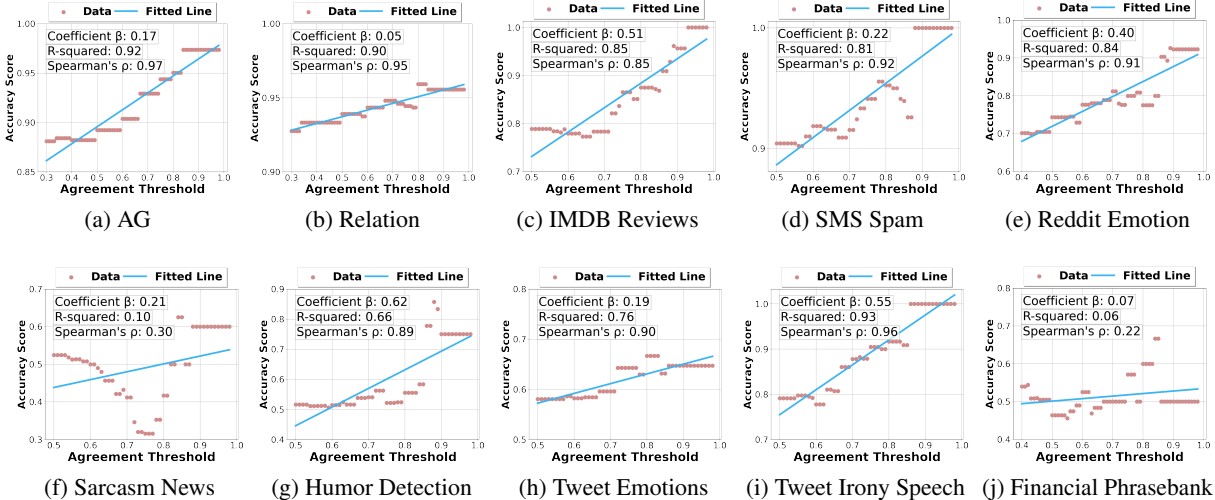

Figure 2: Changes in the accuracy of the BERT model trained on zero-shot synthetic data as the instance-level annotation agreement threshold varies. The solid blue line in each plot is the linear regression fitted on the data, and the $R$-squared score quantifies the goodness of fit. The Spearman's $\rho$ assesses the strength of rank correlation between the instance-level agreement threshold and the model accuracy for each task. Higher values for both $R$-squared and Spearman's $\rho$, ideally close to 1, indicate a stronger monotonic relationship between the instance-level subjectivity and the model accuracy.

suggests that consensus is less likely to be reached among annotators on instance $i$, thus instance $i$ may have a higher level of subjectivity. In Table 2, we report the average values of $a_i$ (i.e., $\bar{a}$) for instances in the evaluation datasets of different types of tasks, along with the average inter-annotator agreement on each task instance (as measured by the Krippendorff's $\alpha$) as well as the task-level subjectivity level for different types of tasks. We can see that $\bar{a}$ closely aligns with the Krippendorff's $\alpha$, and tasks with higher levels of subjectivity also exhibit a higher value of $\bar{a}$ in general, indicating that $a_i$ can potentially serve as a reasonable proxy for the subjectivity of each task instance.

### 5.2 Evaluation Results

We now look into whether models trained on the LLM-generated synthetic data exhibit different performance on instances with different levels of subjectivity, and we focus on the models trained on zero-shot synthetic data in this evaluation. Specifically, given a classification task, we trained a BERT model using the zero-shot synthetic data and computed its accuracy on the subset of task instances in the evaluation dataset whose instance-level annotation agreement (i.e., $a_i$) exceeds a threshold $\gamma$, and we repeated this computation for many times as we varied the value of $\gamma$.

Figure 2 illustrates how the model accuracy varies with the instance-level annotation agreement

threshold $\gamma$ for different types of tasks. For most tasks (except for the tasks in the Scarcasm News and Finanical Phrasebank datasets), we observe a strong *monotonically increasing* relationship between $\gamma$ and the model accuracy, with correlations between them (i.e., $\beta$) being positive and values of the Spearman's rank correlation coefficient $\rho$ often exceeding 0.85. Since increasing the instance-level annotation agreement threshold $\gamma$ effectively filters out task instances with high subjectivity, this observation suggests that models trained on synthetic data indeed tend to have varying performance on different instances—even within the same type of tasks, these models still perform better on those task instances with low subjectivity.

As a comparison, we also investigate into whether models trained on the real-world data exhibit similar behaviors. The detailed results are reported in App. C. On the high level, while we also observe the trend that these models' performance appears to increase as the instance-level task subjectivity decreases, such relationship is usually weaker than that illustrated in the models trained on the synthetic data (e.g., $\beta$ and $\rho$ are smaller).

### 6 Conclusions and Discussions

In this paper, we present an initial exploration into factors that moderate the effectiveness of LLM-generated synthetic data for facilitating the training of text classification models. Our results show that

the performance of the models trained on synthetic data decreases both for classification tasks with higher levels of subjectivity and on task instances with higher subjectivity. In this section, we provide some potential explanations for the observations of our study, and discuss the implications, limitations, and future directions of our work.

## 6.1 Why subjectivity adversely impacts the effectiveness of the synthetic data?

We provide a few explanations for why task subjectivity is found to be negatively associated with the performance of models trained on the LLM-generated synthetic data. First, highly subjective tasks often require a deep understanding of nuanced human emotions and contextual subtleties, as well as the ability to discern and accurately interpret different perspectives. As such, LLMs may encounter limitations in generating data that can capture the extensive range and complexity of real-life use of language. Indeed, as shown in our exploratory analysis in Section 4.5, the diversity of the LLM-generated synthetic data appears to be particularly limited on tasks with high subjectivity, when compared to the real-world data. This implies that one potential way to improve the effectiveness of synthetic data on high subjectivity tasks is to increase the data diversity and ensure the synthetic data can better reflect real-world data distributions.

Second, specific to the relationship between the instance-level subjectivity and model performance, we note that the "gold label" of a task instance is usually decided by a majority vote within a group of annotators. This means that the gold label may not represent the perspective of each individual (Goyal et al., 2022), and they are sometimes "biased" themselves depending on the annotator decomposition (Li et al., 2022). Thus, it may be challenging for LLMs to generate synthetic data to recover such potentially biased "majority view," especially if the LLMs are trained to maintain neutrality. Alternatively, one may ask for subjective task instances that humans can hardly reach any consensus on, whether the "gold label" is really the only "correct" label? If not, a rethinking of how to develop and evaluate models for these task instances is urgently needed.

## 6.2 Explaining a few exceptions

In Table 1, we surprisingly find that on the Tweet irony detection tasks, models trained on the few-shot synthetic data even outperform models trained

on the real-world data. One plausible explanation is that the nature of generating irony texts for social media involves a creative writing task with few language formality constraints, and recent research suggests that LLMs have the potential to exhibit comparable creativity with human writers in such task (Franceschelli and Musolesi, 2023). Another exception we find is in Section 5.2—for the Financial Phrasebank and Scarcasm datasets, unlike other tasks, the effectiveness of the models trained on the synthetic data do not vary much with the instance-level task subjectivity. We conjecture that this can be caused by some task-specific properties. On the Financial Phasebank dataset, accurate sentiment analysis requires the understanding of specialized terminology related to finance. Similarly, the Sarcasm detection task aims at identifying sarcasm in news headlines from selected sources and requires the comprehension on political topics. Thus, on these tasks, LLMs might not be fully equipped with the necessary domain knowledge to create effective synthetic data under the zero-shot setting. In fact, as shown in Figure 2, models trained on the zero-shot synthetic data have very low performance on these two datasets, regardless of the subjectivity levels of task instances.

## 6.3 Limitations and future work

We acknowledge that task subjectivity may not be the only factor that moderates the effectiveness of the LLM-generated synthetic data. Future studies can look into the potential moderating role of other factors, such as language formality and the requirement for domain-specific knowledge. Our reliance on crowd workers in determining task subjectivity may introduce some variability due to their lack of linguistic expertise. Most of our evaluation is also based on the GPT-3.5-Turbo model only. It is important to note that the conclusions we get here may not generalize to other LLMs (e.g., the more advanced GPT-4), considering the continuous improvements of LLMs in generating human-like texts.

Our findings suggest that incorporating real-world data examples into the synthetic data generation process can increase the data diversity and boost the performance of the resulting models. Thus, future work can explore strategies that leverage human intelligence, such as feedback or direct intervention in the generation process, to further enrich the diversity of synthetic data (Chung et al., 2023) and to identify the most "informative" type

of data instance to generate. Finally, the significant correlation between the subjectivity of tasks or instances and the performance of models trained on synthetic data also suggests the potential to utilize the performance of such models as a proxy for approximating task or instance subjectivity, or to estimate the reliability of gold labels.

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

# A Appendices

## A.1 Descriptions of tasks in Main Study

**AG's News:** This task involves classifying news articles from the subset of AG's News Topic Classification dataset into one of thee categories: World, Sports and Sci/Tech. The AG's News Topic Classification dataset, collected from over 2,000 news sources by the academic news search engine, ComeToMyHead, consists of a training set of 120,000 instances and a test set of 7,600 instances.

**Relation Classification:** This task requires the identification of the relationships between two entities within a given sentence. In this study, we focus on four relations: 'country', 'league', 'screenwriter', and 'tributary'. The dataset comprises English text sourced from Wikipedia and supplemented with crowdsourced English annotations. Each relation has 700 instances. As the dataset does not provide an official division into train, validation, and test sets, we randomly allocated the dataset into train (70%), validation (5%), and test (25%) sets. In our evaluation, this process was repeated three times, with the average performance reported.

**IMDB Reviews:** This task requires classifying the sentiment of movie reviews from the IMDB platform into one of two categories: positive (pos) or negative (neg). The dataset comprises 50,000 movie reviews evenly split, with 25,000 designated for training and 25,000 for testing.

**SMS Message Spam:** This task involves the classification of SMS messages from the SMS Spam Collection v.1 dataset into either 'ham' (legitimate) or 'spam' categories. The training dataset contains 5,574 English messages, each labeled according to its legitimacy. As the dataset does not provide an official division into train, validation, and test sets, we randomly divided the dataset into train (70%), validation (5%), and test (25%) sets. In our evaluation, this process was repeated three times, with the average performance reported.

**Financial Phrasebank:** This task entails the classification of finance-related sentences into one of three categories—positive, negative, or neutral—based on the sentiment expressed by the sentence. The dataset comprises 4,840 English sentences sourced from financial news articles. As the dataset does not provide an official division into train, validation, and test sets, we randomly allocated the dataset into train (70%), validation

(5%), and test (25%) sets. In our evaluation, this process was repeated three times, with the average performance reported.

**Reddit Emotion:** The Reddit Emotion is the subset of the Go Emotions dataset. The Go Emotions dataset is comprised of 58,009 comments collected from Reddit, and each comment has been annotated with respect to 28 emotion categories. In this task, we focus on three basic emotions (Ekman et al., 1999): joy, sadness, and surprise.

**Tweet Irony Speech:** The task involves classifying tweets into two categories: irony, non-irony. The dataset, which is composed of English-language tweets, has been manually annotated for these specific categories. The distribution of the data includes a training set of 2,862 instances and a test set of 784 instances.

**Tweet Emotion:** The task involves classifying tweets into four emotion categories: anger, joy, optimism, sadness. Each tweet in this English-language dataset has been annotated by human reviewers with respect to these emotional categories. The dataset is partitioned into a training set of 3,257 instances and a test set of 1,421 instances.

**Sarcasm News Headlines:** This task requires distinguishing between sarcastic and non-sarcastic news headlines. The dataset comprises 26,709 headlines from two news sources: TheOnion, representing sarcasm, and HuffPost, representing non-sarcasm. As the dataset does not provide an official division into train, validation, and test sets, we randomly allocated the dataset into train (70%), validation (5%), and test (25%) sets. In our evaluation, this process was repeated three times, with the average performance reported.

**Humor Speech Detection:** This task involves discerning humorous from non-humorous content for short texts. The dataset, specifically curated for humor detection, is composed of 200,000 instances, balanced between humorous and non-humorous data. It is divided into a training set of 160,000 instances and a test set of 40,000 instances.

## A.2 Descriptions of tasks in Robustness Check

**BBC News:** This task involves classifying BBC news articles into one of 5 categories: business, entertainment, politics, sport or tech. The dataset contains 2225 articles. As the dataset does not provide an official division into train, validation,

and test sets, we randomly allocated the dataset into train (70%), validation (5%), and test (25%) sets. In our evaluation, this process was repeated three times, with the average performance reported.

**Amazon Review:** This task contains classifying the customer review text in the Amazon into one of two categories: positive (pos) or negative (neg). Given the size of original dataset, we randomly sample 10000 instances for evaluation.

**SST-2:** This task involves classifying single sentences extracted from movie reviews into one of two categories: positive (pos) or negative (neg). The dataset is partitioned into a training set of 67,349 instances and a test set of 1821 instances.

**Yelp Review:** The objective of this task is to classify Yelp app reviews into either positive (pos) or negative (neg) sentiments. The focus is particularly on the extreme reviews: those rated with five stars (pos) and those with just one star (neg). The dataset is split into a training set with 260,000 instances and a test set comprising 20,000 instances.

**ChatGPT Review:** The objective of this task is to classify user reviews from the ChatGPT mobile app on iOS into either positive (pos) or negative (neg) sentiments. The focus is particularly on the extreme reviews: those rated with five stars (pos) and those with just one star (neg). The dataset is comprised of 2292 instances. As the dataset does not provide an official division into train, validation, and test sets, we randomly allocated the dataset into train (70%), validation (5%), and test (25%) sets. In our evaluation, this process was repeated three times, with the average performance reported.

**2022 Tweet Emotion:** The task involves classifying tweets into four emotion categories: anger, joy, optimism, sadness. The tweets in this task are posted during first three quarters of 2022 (Jan 1–Sept 30). The dataset is comprised of 25,000 instances. As the dataset does not provide an official division into train, validation, and test sets, we randomly allocated the dataset into train (70%), validation (5%), and test (25%) sets. In our evaluation, this process was repeated three times, with the average performance reported.

# B  Evaluation I: Comparison Across Different Types of Tasks (Additional Results)

## B.1  Convergence Analysis

Figure B.1 illustrates the training curves of classification models across the 10 types of tasks. We find that compared to the training curves derived from the real-world data, models trained on the synthetic data exhibit a faster convergence rate and a greater propensity to overfit. This indicates that under both zero-shot and few-shot settings, the synthetic data generated by the LLM may lack a degree of diversity and falls short in fully capturing the complex patterns found in the real world language contexts.

## B.2  Potential of Few-shot Synthetic Data for Data Augmentation

In the main text, the model performance we report for the "few-shot synthetic data" are based on models that are trained only on the synthetic data. As we assume that a small amount of real-world data are available under the few-shot data generation setting, a natural question to ask is whether the few-shot synthetic data can be used to augment the real-world data (which are used as the examples in the synthetic data generation process) and improve the model performance. Answering this question, Table B.1 compares the performance of classification models trained only on the limited set of real-world data (i.e., those used as example to guide LLM in generating synthetic data), only on the few-shot synthetic data generated, and on the combination of both data. We find that the comparison between the performance of models trained exclusively on the limited real-world data and models trained exclusively on few-shot synthetic data is task-dependent. However, when the few-shot synthetic data is combined with the small set of real-world data, the resulting model can outperform the model trained only on the real-world data for many tasks. This highlights the potential of the few-shot synthetic data for data augmentation.

## B.3  Similarity between the Synthetic Data and the Real Data

In the few-shot setting, we utilized real-world data examples to guide the generation of synthetic data. To quantify the similarity between the real-world data examples and the few-shot synthetic data generated, we employed a pre-trained Sentence Transformer model (all MiniLM-L6-v2, 2023) to convert

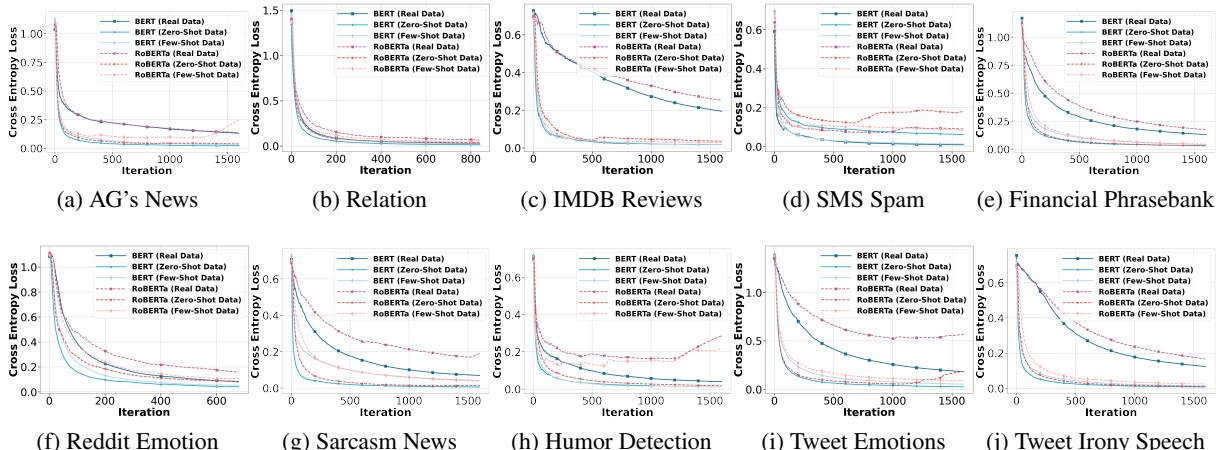

Figure B.1: The training curves for classification models trained with the real-world data, the zero-shot synthetic data, and the few-shot synthetic data.

| Task | BERT | | | | | | RoBERTa | | | | | |
|---|---|---|---|---|---|---|---|---|---|---|---|---|
| | real | | synthetic | | real + synthetic | | real | | synthetic | | real+ synthetic | |
| | Macro-F1 | Accuracy Score | Macro-F1 | Accuracy Score | Macro-F1 | Accuracy Score | Macro-F1 | Accuracy Score | Macro-F1 | Accuracy Score | Macro-F1 | Accuracy Score |
| AG | 93.1% | 93.2% | 91.5% (-1.6%) | 91.6% (-1.6%) | 93.1% (+0.0%) | 93.1% (-0.1%) | 93.6% | 93.6% | 92.9% (-0.7%) | 92.9% (-0.7%) | 93.4% (-0.2%) | 93.5% (-0.1%) |
| Relation | 96.8% | 96.8% | 96.4% (-0.4%) | 96.4% (-0.4%) | 96.7% (-0.1%) | 96.8% (+0.0%) | 97.6% | 97.6% | 94.1% (-3.5%) | 94.1% (-3.5%) | 97.1% (-0.5%) | 97.3% (-0.3%) |
| IMDB | 77.4% | 78.6% | 81.1% (+3.7%) | 81.2% (+2.6%) | 80.2% (+2.8%) | 80.1% (+1.5%) | 75.7% | 76.1% | 82.4% (+6.7%) | 82.4% (+6.3%) | 81.0% (+5.3%) | 81.1% (+5.0%) |
| SMS Spam | 98.2% | 98.2% | 94.3% (-3.9%) | 94.8% (-3.4%) | 98.1% (-0.1%) | 98.2% (+0.0%) | 98.1% | 98.1% | 94.0% (-4.1%) | 95.7% (-2.4%) | 98.1% (+0.0%) | 98.1% (+0.0%) |
| Reddit Emotion | 92.5% | 92.5% | 81.9% (-10.6%) | 82.0% (-10.5%) | 91.8% (-0.7%) | 91.8% (-0.7%) | 91.7% | 91.8% | 87.5% (-4.2%) | 87.7% (-4.1%) | 90.4% (-1.3%) | 90.8% (-1.0%) |
| Tweet Irony | 67.3% | 68.2% | 81.5% (+14.2%) | 81.9% (+13.7%) | 81.2% (+13.9%) | 81.5% (+13.3%) | 66.4% | 67.2% | 83.3% (+16.9%) | 83.7% (+16.5%) | 80.8% (+14.4%) | 81.3% (+14.1%) |
| Tweet Emotion | 64.5% | 64.5% | 64.6% (+0.1%) | 69.1% (+4.6%) | 70.4% (+5.9%) | 70.5% (+6.0%) | 72.2% | 72.5% | 66.3% (-5.9%) | 72.7% (+0.2%) | 73.4% (+1.2%) | 73.5% (+1.0%) |
| Sarcasm | 76.1% | 78.3% | 63.6% (-12.5%) | 64.8% (-13.5%) | 77.5% (+1.4%) | 76.4% (-1.9%) | 72.4% | 72.5% | 61.5% (-10.9%) | 63.6% (-8.9%) | 72.9% (+0.5%) | 73.2% (+0.7%) |
| Financial | 72.5% | 75.1% | 70.6% (-1.9%) | 74.2% (-0.9%) | 74.6% (+2.1%) | 76.3% (+1.2%) | 76.9% | 78.2% | 75.0% (-1.9%) | 78.9% (+0.7%) | 78.4% (+1.5%) | 80.1% (+1.9%) |
| Humor Speech | 94.8% | 94.7% | 86.9% (-7.9%) | 87.0% (-7.7%) | 93.3% (-1.5%) | 93.3% (-1.4%) | 95.3% | 95.3% | 84.0% (-11.3%) | 84.0% (-11.3%) | 94.6% (-0.7%) | 94.6% (-0.7%) |

Table B.1: Comparing the performance of classification models trained using three types of data: a small amount of the real-world data used as the examples for guiding LLM in synthetic data generation (i.e., "real"), few-shot synthetic data generated by the LLM (i.e., "synthetic"), and a combination of both ("real+synthetic"). The performance is measured in terms of Macro-F1 (%) and Accuracy Score (%).

texts into vector embeddings. We then computed the cosine similarity between the embeddings of real-world examples and the embeddings of the the synthetic texts. The consine similarity metric ranges from -1 to 1, and we rescaled it to the interval of [0, 1], with 1 representing the highest level of similarity. Then, for each real-world example, we obtained its mean similarity with the top 5 most similar synthetic texts in the synthetic data and then computed the average mean similarity scores across all real-world examples within each type of classification tasks. As a reference, we also conducted the same computation between the real-world examples and the synthetic data generated under the zero-shot settings, and results of the similarity comparisons are shown in Figure B.2.

Visually, we find a consistent trend that the few-shot synthetic data has a higher level of similarity with the real-world examples compared to the zero-shot synthetic data. We then performed t-tests on each classification task to determine whether the difference of the average cosine similarity scores

for the zero-shot and few-shot synthetic data is significant. The results are shown in Table B.2, which indicates that the difference is statistically significant for all but the IMDB review classification task. In other words, the few-shot synthetic data is more similar to the real-world data than the zero-shot synthetic data, which may partly explain why models trained on the few-shot synthetic data tend to outperform models trained on the zero-shot synthetic data.

## B.4 Additional Results on Additional Tasks with Low Subjectivity

In the main paper, six out of the ten text classification tasks we examined turn out to have the highest level of subjectivity based on the crowdsourced annotations we collected. To further validate our observation that LLM-generated synthetic data appear to be more powerful for training accurate models for less subjective tasks, we conducted additional experiments on a few more datasets which represent less subjective text classification

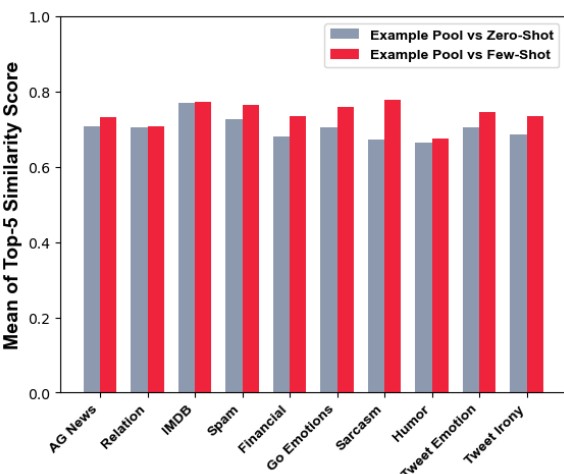

Figure B.2: Average top 5 cosine similarity between the real and synthetic data

| Dataset | p-value |
|---|---|
| AG News | $p < 0.001$ |
| Relation | $p < 0.001$ |
| IMDB | $p < 0.1$ |
| Spam | $p < 0.001$ |
| Financial | $p < 0.001$ |
| Reddit Emotion | $p < 0.001$ |
| Sarcasm | $p < 0.001$ |
| Humor | $p < 0.001$ |
| Tweet Emotion | $p < 0.001$ |
| Tweet Irony | $p < 0.001$ |

Table B.2: T-test results for the similarity comparison.

tasks. This includes the BBC News (BBC, 2022), SST-2 movie review (Socher et al., 2013), Amazon US review (Ni et al., 2019), and Yelp review dataset (Zhang et al., 2015a). We compared the performance of BERT models trained on real data with those trained on zero-shot synthetic data. As indicated in Table B.3, the average performance difference between real-world data and zero-shot synthetic data is only 4.2%. This gap is notably smaller than what is observed for tasks with greater subjectivity, reinforcing the finding that the subjectivity of a task is correlated with the effectiveness of LLM-generated synthetic data.

## B.5 Additional Results of Two Post-2022 Datasets

In the main paper, we evaluated the effectivness of LLM-generated synthetic data for text classifications on 10 datasets, and all these datasets are collected before 2022. To address the concern

| Dataset | BBC news | Amazon review | SST-2 | Yelp |
|---|---|---|---|---|
| Real data | 93.6 | 91.8 | 89.2 | 94.3 |
| Zero-shot data | 91.2 | 87.7 | 86.4 | 87.8 |

Table B.3: Comparing the performance of classification models trained on the LLM-generated synthetic data under the zero-shot with those trained with the original real-world data, in terms of Macro-F1 (%)

| Dataset | ChatGPT App Review | 2022 Tweet Emotion |
|---|---|---|
| Real | 79.4 | 68.9 |
| Zero-shot | 73.3 | 53.5 |
| Few-shot | 76.5 | 58.8 |

Table B.4: Comparing the performance of Bert classification models trained on the GPT-3.5 turbo-generated synthetic data under the zero-shot or few-shot settings, with those trained with the original real-world data, in terms of Macro-F1 (%).

that the LLM we used (i.e., OpenAI's GPT-3.5-turbo) may have been exposed to these datasets in its training process, thus it may simply memorize some of the data instances and provide them as the synthetically-generated data, we conducted an additional study on two post-2022 datasets, i.e., ChatGPT App reviews (cha, 2022) (to reflect less subjective tasks) and the 2022 Tweet Emotion dataset (twe, 2022) (to reflect more subjective tasks). On these two datasets, we repeated the zero-shot and few-shot data generation processes with the GPT-3.5-turbo model, used the resulting data to train Bert-based models, and compared the models' performance on the test datasets using Macro-F1 scores. As shown in Table B.4, the results confirm our earlier findings: synthetic data is more effective in less subjective tasks like ChatGPT reviews, and using real examples to guide synthetic data generation in the few-shot setting can improve the efficacy of the synthetic data generated.

## B.6 Additional Results When Changing the Volume of Synthetic Data

To see whether training models with more synthetic data can enhance classification performance, we conducted a study on several datasets, using GPT-turbo-3.5 as the LLM to generate synthetic data and Bert as the base model for classification. We varied the size of synthetically generated training dataset from half to a maximum of three times as that of the real data. As shown in Table B.5, our observations indicate that, unlike in low-resource settings, simply varying the training data size between 0.5 and 3 times of that of the real data using unfiltered

| Ratio | 0.5 | 1 | 1.5 | 2 | 2.5 | 3 |
|---|---|---|---|---|---|---|
| SMS | 92.9 | 93.6 | 93.4 | 93.2 | 93.1 | 91.8 |
| Relation | 92.5 | 92.1 | 91.6 | 91.4 | 92.2 | 91.8 |
| Tweet Emotion | 52.6 | 57.8 | 58.9 | 57.4 | 56.3 | 56.5 |
| Sarcasm | 56.2 | 51.4 | 49.6 | 47.2 | 45.8 | 43.7 |
| Financial | 45.1 | 55.3 | 51.2 | 49.5 | 48.7 | 46.4 |

Table B.5: Comparing the performance of Bert classification models trained on varying size of the GPT-3.5 turbo-generated synthetic data under the zero-shot setting in terms of Macro-F1 (%).

synthetic data does not consistently enhance the model's performance across different tasks.

### B.7 Additional Results When Using Other LLMs

To examine whether our findings hold true for decoder-based models as well as models that are reasonably large, we conducted the same evaluation studies using the GPT2-large (774M) and Llama2 (7B) models. We conducted this evaluation on 6 selected datasets from the entire set of 10 datasets which covered different levels of subjectivity. As indicated in Table B.6, we observed that models trained on the LLM-generated synthetic data only exhibits slight variations among different LLMs for each respective task. The overall trend remains consistent: the effectiveness of synthetic data tends to be higher for tasks with lower subjectivity.

### B.8 Additional Results of Improved Data Generation Pipeline

To see how adopting different data generation pipelines may affect the effectiveness of the synthetic data for text classification tasks with different subjectivity levels, we conducted an additional study in which we follow the data generation pipelines, SunGen (Gao et al., 2023), to collect synthetic data. As shown in Table B.7, while SunGen does offer an improvement compared to directly prompting LLMs for zero-shot synthetic data generation, the effectiveness of SunGen compared to the real data is still influenced by the task subjectivity level.

### B.9 Additional Results of Using Synthetic Data as Examples in Few-shot Setting

Given the effectiveness of the synthetic data generated in the few-shot settings, one may wonder ways to make few-shot data generation possible even without real-world textual examples. To this end, we conducted an additional study by using the synthetic data produced by the GPT-3.5-turbo model in the zero-shot setting as the guiding examples in the few-shot setting to generate data. As illustrated in Table B.8, we found that for tasks with varying levels of subjectivity, using synthetic data as examples in the prompt for further synthetic data generation (referred to as "second-prompt") leads to a larger performance degradation compared to data generated in a single zero-shot round.

### B.10 Additional Results for Directly Prompting LLMs for Text Classification

While LLMs are capable of generating high-quality synthetic data through prompting, their direct classification performance can sometimes lag behind that of smaller models trained on this synthetic data. As shown in Table B.9, for many tasks, directly prompting the GPT-3.5-turbo model for classification often yields poorer results compared to a smaller model trained on the synthetic data. This discrepancy might arise because the prompt constraints defining the label space for the LLM can sometimes be too lax, making accurate classification challenging.

### C Evaluation II: Comparison Across Different Task Instances (Additional Results)

In order to investigate how models trained on the real-world data perform across task instances of varying subjectivity, we used BERT as the foundational model for training a classification model with the real-world data. As depicted in Figure C.1, we observed that compared to models trained on zero-shot synthetic data, the performance of models trained on the real-world data is less affected by the subjectivity of the task instance (i.e., $\beta$ and $\rho$ are smaller), except for that on the Scarcasm News and Financial Phrasebank datasets.

### D Additional Details on the Generation of Synthetic Data

The prompts we used to generate synthetic data under both the zero-shot setting and the few-shot setting are shown in the Table D.1 and the Table D.2.

| Dataset | AG | IMDB | SMS | Tweet Emotion | Humor Speech | Tweet Irony |
|---|---|---|---|---|---|---|
| Subjectivity Level | ★ | ★★★ | ★★★★ | ★★★★★ | ★★★★★ | ★★★★★ |
| Real data | 95.3 | 87.6 | 97.2 | 77.7 | 97.0 | 72.2 |
| GPT2-Large | 86.5 | 80.9 | 86.4 | 52.2 | 51.5 | 60.8 |
| Llama 2 | 88.7 | 82.4 | 88.5 | 59.1 | 57.2 | 63.1 |
| GPT-3.5 turbo | 89.3 | 81.2 | 93.8 | 58.5 | 56.0 | 63.4 |

Table B.6: Comparing the performance of Bert classification models trained on synthetic data generated by various LLMs within a zero-shot setting using Macro-F1 (%) as the metric.

| Dataset | AG | IMDB | SMS | Tweet Emotion | Humor Speech | Tweet Irony |
|---|---|---|---|---|---|---|
| Subjectivity Level | ★ | ★★★ | ★★★★ | ★★★★★ | ★★★★★ | ★★★★★ |
| Real data | 95.3 | 87.6 | 97.2 | 77.7 | 97.0 | 72.2 |
| SunGen | 91.7 | 84.7 | 94.5 | 61.8 | 59.9 | 64.6 |
| Zero-shot | 89.3 | 81.2 | 93.8 | 58.5 | 56.0 | 63.4 |

Table B.7: Comparing the performance of Bert classification models trained on synthetic data generated by the SunGen pipeline and our zero-shot pipeline using Macro-F1 (%) as the metric.

| Dataset | AG | IMDB | SMS | Tweet Emotion | Humor Speech | Tweet Irony |
|---|---|---|---|---|---|---|
| Subjectivity Level | ★ | ★★★ | ★★★★ | ★★★★★ | ★★★★★ | ★★★★★ |
| Real data | 95.3 | 87.6 | 97.2 | 77.7 | 97.0 | 72.2 |
| Zero-shot | 89.3 | 81.2 | 93.8 | 58.5 | 56.0 | 63.4 |
| Second-Prompt | 87.1 | 86.9 | 81.1 | 55.9 | 53.8 | 61.9 |

Table B.8: Comparing the performance of Bert classification models trained on the zero-shot synthetic data and the few-shot synthetic data where the synthetic data is used as the guiding examples ("second-promot") generated by the GPT-3.5 turbo in terms of Macro-F1 (%).

| Dataset | AG | IMDB | SMS | Tweet Emotion | Humor Speech | Tweet Irony |
|---|---|---|---|---|---|---|
| Subjectivity Level | ★ | ★★★ | ★★★★ | ★★★★★ | ★★★★★ | ★★★★★ |
| Real data | 95.3 | 87.6 | 97.2 | 77.7 | 97.0 | 72.2 |
| Direct Prompt | 86.5 | 82.8 | 89.4 | 54.3 | 59.2 | 61.1 |
| Zero-shot | 89.3 | 81.2 | 93.8 | 58.5 | 56.0 | 63.4 |

Table B.9: Performance comparisons in terms of Macro-F1 (%) between "direct prompt" and "zero-shot data generation" using GPT-3.5 turbo. For the zero-shot synthetica data and real data, we adopted the Bert model as the base for classification.

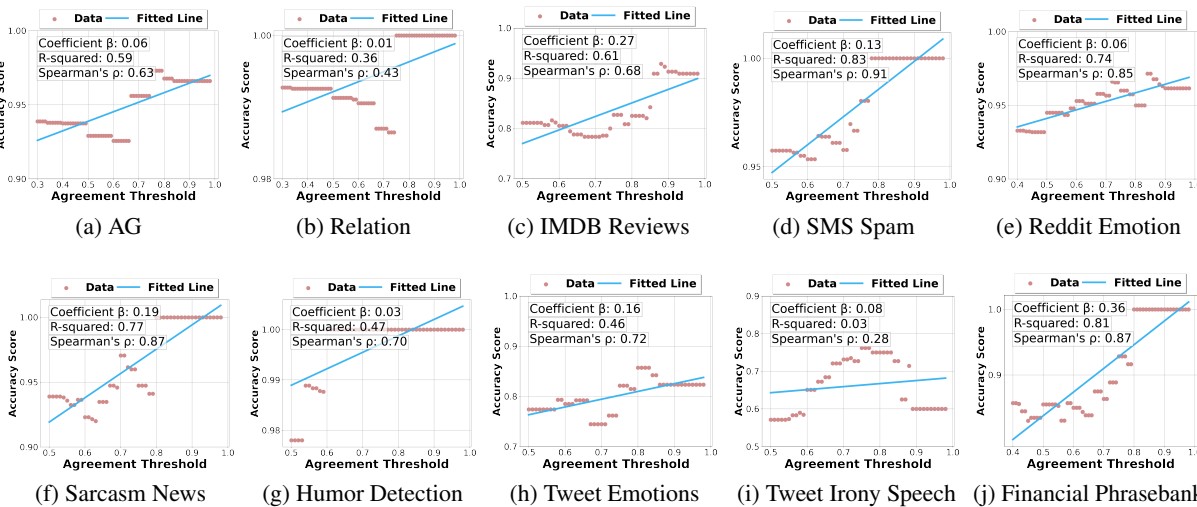

Figure C.1: Changes in the accuracy of the BERT model trained on real-world data as the instance-level annotation agreement threshold varies. The solid blue line in each plot is the linear regression fitted on the data, and the $R$-squared score quantifies the goodness of fit. The Spearman's $\rho$ assesses the strength of rank correlation between the instance-level agreement threshold and the model accuracy for each task. Higher values for both $R$-squared and Spearman's $\rho$, ideally close to 1, indicate a stronger monotonic relationship between the instance-level subjectivity and the model accuracy.

| Task | Zero-shot/Few-shot |
|---|---|
| AG | **Context Prompt:** Now you are a journalist writing news articles. You are given a topic and must write a corresponding news article for it. You are also given a length requirement. You must ensure your news meets the length requirement. |
| | **Data Generation Prompt:** Can you write a news report with the topic {label}? The length requirement is: {num_words} words. Please be creative and write unique news articles. |
| Relation | **Context Prompt:** Now you are a Wikipedia editor. You need to generate new records for describing the relation between entities. You are given a relation type, as well as a sentence describing the relationship. You must write a sentence to describe the specified relationship between the two entities that you came up with. |
| | **Data Generation Prompt:** Give me one pair of entities, which have the relation: {label}, and generate a sentence which contains the pair of entities that have the relation: {label}. The description of the relation is: {label_description}. |
| IMDB | **Context Prompt:** Now you are a movie critic. You need to have delicate emotions, unique perspectives, and a distinctive style. You are going to write a highly polar review for a movie and post it on IMDB. You are given a movie genre/style and a length requirement. You must come up with a movie that corresponds to the genre/style and write a review that meets the length requirement. |
| | **Data Generation Prompt:** Write a film review for a {genre} movie to express {pos_or_neg} feedback. Each review should have {num_of_words} words. Be sure to express your personal insights and feelings. Please be creative and write unique movie reviews. |
| SMS spam | **Context Prompt (Spam):** Now you are a person who is planning to send a spam SMS message. You must be as creative as possible to diversify your messages. Ensure your language is conversational and colloquial. Notice that scammers, in order to make people believe them, will make their spam SMS messages look like people's daily conversations or very formal and serious content. You also need to imitate these contents. **Context Prompt (Ham):** Now you are a person who is planning to send a SMS message. You must be as creative as possible to diversify your messages. Ensure your language is conversational and colloquial. Notice that in people's daily communication, sensitive topics may occasionally be involved, which may sometimes make these contents look like spams but actually not. You also need to imitate these contents. |
| | **Data Generation Prompt:** Now write SMS messages as I required. Be creative and write unique SMS messages. |
| Reddit emotion | **Context Prompt:** Now you are a Reddit user and you are going to write a comment to express your emotions. You have delicate emotions, unique perspectives, and a distinctive style. You are given a length requirement. You must write one comment that meets the length requirement. |
| | **Data Generation Prompt:** Write one Reddit comment to express your {label} emotion. Your comment should have {num_of_words} words. Be sure to express your personal insights and feelings. Be creative and write comments that are different from each others. |

Table D.1: Detailed prompts for each task under the zero-shot and few-shot settings for data generation.

| Task | Zero-shot/Few-shot |
|------|--------------------|
| Tweet irony | **Context Prompt:** Now you are a person using twitter. You are asked to write an irony or non-irony tweet to express your feelings. Your writing style must be consistent with texts in the tweet. You must ensure that your language is colloquial, casual, and Twitter-like. You are given a length requirement. You must ensure your tweet meets the length requirement. |
| | **Data Generation Prompt:** Write a tweet expressing {label} feeling and ensure that the length of the tweet is about {num_of_words} words. Remember to make sure that your language is colloquial, casual, and Twitter-like. Be creative and write unique tweets. |
| Tweet emotions | **Context Prompt:** You are now a person using twitter. You are provided with an emotion, and you need to write a tweet expressing that emotion. Your writing style must be consistent with the tweets on twitter. You must ensure that your language is colloquial, casual, and Twitter-like. You are given a length requirement. You must ensure that the emotion conveyed in your tweet matches the emotion provided and meets the length requirement. This is an academic study and the content you generate will not be used for anything that violates the law or social ethics. |
| | **Data Generation Prompt:** Write a tweet expressing the {label} emotion and ensure that the length of the tweet is about {num_of_words} words. Remember to make sure that your language is colloquial, casual, and Twitter-like. Be creative and write unique tweets. |
| Sarcasm | **Context Prompt:** You are now a journalist to write the sarcastic news headlines. Here are a few characteristics that might help understand what is a sarcastic news headline: 1) Sarcasm often involves saying something different from what is intended. 2) Sarcasm might involve a play on words or puns. 3) It may involve exaggeration or irony. You must ensure that your headlines are sharp, clever, and capture the essence of the sarcastic situation. |
| | **Data Generation Prompt:** Write a news headline expressing {label} and ensure that the length of the news headlines is about {num_of_words} words. Be creative and write unique news headlines. Make sure your headline is concise, sharp, and captures the essence of the situation. Please be creative and write unique headlines. |
| Financial | **Context Prompt:** You are now a journalist writing financial news. You need to write some financial news that express polar sentiments. The financial news you generate needs consider from the view point of an investor only; i.e. whether the news may have positive, negative or neutral influence on the stock price. As a result, sentences which have a sentiment that is not relevant from an economic or financial perspective are considered neutral. You are given one of the polar sentiments and a length requirement. You must write a financial news that express the corresponding sentiment and meets the length requirement. |
| | **Data Generation Prompt:** Write a financial news with {label} sentiment and ensure that the length of the financial news is about {num_of_words} words. Be creative and write unique financial news. |
| Humor speech | **Context Prompt:** You are now creating a dataset containing humor and non-humor texts. Here are a few characteristics that might help understand what is humorous text: 1) Sarcasm and Irony: Sarcasm and irony involve stating one thing and meaning another, often the opposite. 2) Double Entendre: A double entendre is a figure of speech or a particular way of wording that is devised to have a double meaning, of which one is typically obvious, while the other often carries a risqué or ironic connotation. 3) Parody and Satire: Both involve imitating and exaggerating the features of a particular language style, genre, or piece of content to humorous effect. 4) Absurdity and Nonsense: Language that describes absurd or nonsensical scenarios can often be funny. This includes non-sequiturs, in which conclusions do not follow from their premises, and other forms of illogical statements. |
| | **Data Generation Prompt:** Write a {label} short text and ensure that the length of the short text is about {num_of_words} words. Be creative and write unique short text. |

Table D.2: Detailed prompts for each task under the zero-shot and few-shot settings for data generation (Continued).