# OpenReview forum: "Synthetic Data Generation with Large Language Models for Text Classification: Potential and Limitations"
_EMNLP/2023/Conference — EMNLP 2023 Main_

### Official Review · Reviewer_uLMj · 2023-08-04

**Soundness:** 2

**Excitement:**

4: Strong: This paper deepens the understanding of some phenomenon or lowers the barriers to an existing research direction.

**Paper Topic And Main Contributions:**

This longer paper describes the use LLMs to generate synthetic data for text classification. Experiments show that LLM-generated synthetic data is very close to those trained on the real-world data for tasks with low subjectivity while the performance decrease is much bigger on tasks with high subjectivity (e.g., humor or sarcasm detection).

**Questions For The Authors:**

Section 4.2. Why not also rank datasets on "instance-level subjectivity", as you defined it, using a sample from each dataset and measuring inter-annotator (dis)agreement?  Also, Table 1 subjectivity ranking are, in some cases, counter-intuitive and makes one question the effectiveness of the described crowd-sourcing approach.

Section 6.2 The explanation for synthetic-data training outperforming real data on the dataset Tweet irony is not very convincing. Have you analyzed examples from the real vs the synthetic dataset?

Why not compare results to zero-shot/few-shot classification based on the same LLM, vs synthetic data + trained classification model? Does the LLM still perform better on less subjective datasets?

The more "subjective" (per your measure) datasets in your experiments significantly outnumber the  less "subjective" datasets. It is unclear that the difference in performance is, therefore, really due to the level of subjectivity or due to the data used by the LLM. Similarly, since a single LLM is used to generate synthetic data, the generalizability of the results is questionable.

**Reasons To Accept:**

A very well written and thorough paper.

Results are reproducible.

**Reasons To Reject:**

Both the novelty, the conducted experiments, and meaningful insights are limited.  The paper could be of interest to a small group of researchers and is more suitable for a short paper.

The measurement of datasets by subjectivity (obtained via crowdsourcing) could be questioned. Similarly, the drawn conclusion (LLM synthetic data training on subjective datasets performs worse than on less subjective datasets) could be questioned because of 1) very few "non" subjective datasets used in experiments and 2) a single LLM model.

**Reproducibility:**

4: Could mostly reproduce the results, but there may be some variation because of sample variance or minor variations in their interpretation of the protocol or method.

**Reviewer Confidence:**

4: Quite sure. I tried to check the important points carefully. It's unlikely, though conceivable, that I missed something that should affect my ratings.

---

> ### Author Rebuttal · Authors · 2023-08-29
>
> Thanks for your thoughtful feedback. Below, we address your main questions.
>
> > Section 4.2. Why not also rank datasets on "instance-level subjectivity", as you defined it, using a sample from each dataset and measuring inter-annotator (dis)agreement? Also, Table 1 subjectivity ranking are, in some cases, counter-intuitive and makes one question the effectiveness of the described crowd-sourcing approach.
>
> In Section 4.2, we explore subjectivity at the task level as different instances within the same task can exhibit varying degrees of subjectivity. That said, as illustrated in Table 2 in the original paper, the ranking of task-level subjectivity computed based on the average instance-level disagreement for each type of task closely aligns with the task-level subjectivity that we obtained from our crowdsourced study. We will include the rank based on average instance-level disagreement for each dataset in Table 1 in the original paper to facilitate the understanding.
>
> >Section 6.2 The explanation for synthetic-data training outperforming real data on the dataset Tweet irony is not very convincing. Have you analyzed examples from the real vs the synthetic dataset?
>
>  We observed that the enhanced performance in the few-shot synthetic data primarily stems from a decrease in the false positive rate when classifying the non-irony tweets as irony. Upon examining real tweets in the dataset, a small subset was identified where, despite being labeled as non\_irony, the tweets included hashtags  #irony and #sarcasm. Here are a couple of examples: 1) @user please note #sarcasm in last tweet before you kill yourself ;) many good pros including @user can show you best usage. 2) @user @user republicans suck more at identifying #irony than alanis morissette. In addition, a significant proportion of the tweets labeled as irony in the real data also contain the hashtags #irony and #sarcasm.  When models are trained using real data,  they only encounter a small number of instances that are labeled as non\_irony but contain hashtags like \#irony or \#sarcasm. As a result, when tested, these models exhibit a high false positive rate for those cases. However, in few-shot scenarios, introducing such samples to the LLM effectively augments their presence in the training set by several folds. This augmentation seems to facilitate more accurate predictions for such instances in the test set. This observation aligns with Aggarwal's 2023 study [1], which demonstrated that leveraging a larger volume of synthetic data (2-10x real data) in low-resource situations can improve the performance on real data.
>
> [1] Aggarwal, K., Jin, H. and Ahmad, A., 2023. ECG-QALM: Entity-controlled synthetic text generation using contextual Q&A for NER.
>
> > Why not compare results to zero-shot/few-shot classification based on the same LLM, vs synthetic data + trained classification model? Does the LLM still perform better on less subjective datasets?
>
> Directly applying LLMs for text classification may present several concerns:
>
> a) __Significant Computational Cost__: Directly Using LLMs for text classification demands considerably higher computational resources compared to smaller models trained on the synthetic data. This becomes especially evident in mobile scenarios, where deploying LLMs is often impractical due to inherent resource constraints.
>
> b) __Data Privacy and Security__: In contexts with stringent data security or privacy requirements, organizations may opt for their own smaller, bespoke models rather than relying on external LLM APIs, which they can't directly control or oversee.
>
> c)  __Potential Performance Gap__: While LLMs are capable of generating high-quality synthetic data through prompting, their direct classification performance can sometimes lag behind that of smaller models trained on this synthetic data. As illustrated in Table 1 below,
>
> **Table 1:** Performance comparisons in terms of Macro-F1 (%) between ''direct prompt'' and ''zero-shot data generation'' using GPT-3.5 turbo. For the "zero-shot" setting and "real data", we adopted the Bert model as the base for classification.
>
> |                    | AG   | IMDB | SMS Spam | Tweet Emotion | Humor | Tweet Irony |
> |--------------------|------|------|----------|---------------|-------|-------------|
> | Subjectivity level | *    | ***  | ****     | *****         | ***** | *****       |
> | Real data          | 95.3 | 87.6 | 97.2     | 77.7          | 97.0  | 72.2        |
> | Directly prompt    | 86.5 | 82.8 | 89.4     | 54.3          | 59.2  | 61.1        |
> | Zero-shot          | 89.3 | 81.2 | 93.8     | 58.1          | 56.0  | 63.4        |
>
> for many tasks, directly prompting GPT-3.5 turbo for classification often yields poorer results compared to a smaller model trained on the synthetic data. This discrepancy might arise because the prompt constraints defining the label space for the LLM can sometimes be too lax, making accurate classification challenging.
>
> Furthermore, we have observed that using large language models (LLMs) directly for classification also yields the smaller performance gap between the performance on models trained on real data on tasks with lower subjectivity.
>
> >The more "subjective" (per your measure) datasets in your experiments significantly outnumber the less "subjective" datasets. It is unclear that the difference in performance is, therefore, really due to the level of subjectivity or due to the data used by the LLM. Similarly, since a single LLM is used to generate synthetic data, the generalizability of the results is questionable.
>
> Thanks for your concern.
>
> A) To validate our observations regarding ``subjectivity'' in the data, we conducted additional experiments on a few more datasets which represent less subjective text classification tasks: the BBC News dataset, SST-2 movie review, Amazon Us review , Yelp review, and ChatGPT App review. As depicted in Table 2 below,
>
> **Table 2:** Comparing the performance of classification models trained on the LLM-generated synthetic data under the zero-shot  with those trained with the original real-world data, in terms of Macro-F1 (\%).
>
> | Dataset   | BBC  | Amazon | SST-2 | Yelp | ChatGPT app review |
> |-----------|------|--------|-------|------|--------------------|
> | Real data | 93.6 | 91.8   | 89.2  | 94.3 | 79.4               |
> | Zero-shot | 91.2 | 87.7   | 86.4  | 87.9 | 73.3               |
>
>
> the average performance difference between real-world data and synthetic data is only 4.2%. This gap is notably smaller than what is observed in tasks with greater subjectivity, reinforcing the finding that the subjectivity of a task can indeed diminish the effectiveness of synthetic data.
>
>
> B) To examine whether our findings hold true for decoder-based models as well as models that are reasonably large,  we conducted the same evaluation studies using the GPT2-large (774M) and Llama2 (7B) models.
> Due to the time constraints, we conducted this evaluation on 6 randomly selected  datasets from the entire set of 10 datasets that we considered in our paper, and the selected 6 datasets covered different levels of subjectivity. As indicated in Table 3 below,
>
> **Table 3:** Comparing the performance of Bert classification models trained on synthetic data generated by various LLMs within a zero-shot setting using Macro-F1 (\%) as the metric.
> |                    |  AG  | IMDB |  SMS | Tweet Emotion | Humor Speech | Tweet Irony |
> |:------------------:|:----:|:----:|:----:|:-------------:|:------------:|:-----------:|
> | Subjectivity level |   *  |  *** | **** |     *****     |     *****    |    *****    |
> |      Real Data     | 95.3 | 87.6 | 97.2 |      77.7     |     97.0     |     72.2    |
> |    GPT2-large (774M)    | 86.5 | 80.9 | 86.4 |      52.2     |     51.5     |     60.8    |
> |    Llama 2 (7B)    | 88.9 | 82.4 | 88.5 |      59.1     |     57.2     |     63.1    |
> |    GPT-3.5 turbo   | 89.3 | 81.2 | 93.8 |      58.5     |      56.0      |     63.4    |
>
> we observed that models trained on the LLM-generated synthetic data only exhibits slight variations among different LLMs for each respective task.  The overall trend as seen in our paper remains consistent: the effectiveness of synthetic data tends to be higher for tasks with lower subjectivity.
>
> #### We will add the additional experimental results into the later paper or the appendix.

---

### Official Review · Reviewer_PuaF · 2023-08-06

**Typos Grammar Style And Presentation Improvements:** None found.
**Soundness:** 4

**Excitement:**

3: Ambivalent: It has merits (e.g., it reports state-of-the-art results, the idea is nice), but there are key weaknesses (e.g., it describes incremental work), and it can significantly benefit from another round of revision. However, I won't object to accepting it if my co-reviewers champion it.

**Missing References:**

None

**Paper Topic And Main Contributions:**

The paper demonstrates empirically that more subjective text classification tasks or individual training examples are less effective for synthetic training data generation with Large Language Models.

**Questions For The Authors:**

I only have suggestions what else to consider in future revisions rather than specific questions:

A. How sensitive the results are to the prompts asking LLM to generate training data? I did not see any ablations.

B. It may be interesting to try (or to comment) on a setup with multiple generation chains where synthetic examples are used in the prompt rather than real example to generate more synthetic examples. If the overall quality degrades even further, that will be additional evidence that synthesizing examples can be considered a noisy channel.

**Reasons To Accept:**

Timely topic. The paper is well written and empirical findings are convincing.

**Reasons To Reject:**

1) The setup is a bit artificial: if we believe that LLM (e.g. GPT3) is a good model that is capable of synthesizing training data, then why we only use it for synthesizing but not for the classification directly? The authors use a smaller Bert-size model to perform classification. So, this work may be better positioned as distillation work rather than classification improvement.
2) The findings are not surprizing: more subjective tasks or training examples results in more noise in the synthetic training data.


**Reproducibility:**

5: Could easily reproduce the results.

**Reviewer Confidence:**

4: Quite sure. I tried to check the important points carefully. It's unlikely, though conceivable, that I missed something that should affect my ratings.

---

> ### Author Rebuttal · Authors · 2023-08-29
>
> Thanks for thoughtful review. Below, we address your main questions.
>
> > The setup is a bit artificial: if we believe that LLM (e.g. GPT3) is a good model that is capable of synthesizing training data, then why we only use it for synthesizing but not for the classification directly? The authors use a smaller Bert-size model to perform classification. So, this work may be better positioned as distillation work rather than classification improvement.
>
>  Directly applying LLMs for text classification may present several concerns:
>
> a)  __Significant Computational Cost__: Directly Using LLMs for text classification demands considerably higher computational resources compared to smaller models trained on the synthetic data. This becomes especially evident in mobile scenarios, where deploying LLMs is often impractical due to inherent resource constraints.
>
> b) __Data Privacy and Security__: In contexts with stringent data security or privacy requirements, organizations may opt for their own smaller, bespoke models rather than relying on external LLM APIs, which they can't directly control or oversee.
>
> c) __Potential Performance Gap__: While LLMs are capable of generating high-quality synthetic data through prompting, their direct classification performance can sometimes lag behind that of smaller models trained on this synthetic data. As illustrated in Table1 below,
>
> **Table 1:** Performance comparisons in terms of Macro-F1 (%) between ''direct prompt'' and ''zero-shot data generation'' using GPT-3.5 turbo. For the "zero-shot" setting and "real data", we adopted the Bert model as the base for classification.
>
> |                    | AG   | IMDB | SMS Spam | Tweet Emotion | Humor | Tweet Irony |
> |--------------------|------|------|----------|---------------|-------|-------------|
> | Subjectivity level | *    | ***  | ****     | *****         | ***** | *****       |
> | Real data          | 95.3 | 87.6 | 97.2     | 77.7          | 97.0  | 72.2        |
> | Directly prompt    | 86.5 | 82.8 | 89.4     | 54.3          | 59.2  | 61.1        |
> | Zero-shot          | 89.3 | 81.2 | 93.8     | 58.1          | 56.0  | 63.4        |
>
> for many tasks, directly prompting GPT-3.5  for classification often yields poorer results compared to a smaller model trained on the synthetic data. This discrepancy might arise because the prompt constraints defining the label space for the LLM can sometimes be too lax, making accurate classification challenging.
>
> > How sensitive the results are to the prompts asking LLM to generate training data? I did not see any ablations.
>
> To understand the sensitivity of the prompt used for generating synthetic data in a zero-shot setting, we performed an ablation study based on GPT-3.5 turbo. We remove the context prompt and diversity prompt from our original prompt to form what we term a``bad prompt'' for the data generation. A shown in the Table 2 below,
>
> **Table 2:** Comparing the performance of Bert classification models trained on the GPT-3.5 turbo-generated  zero-shot synthetic data with different prompts in terms of Macro-F1 (\%).
> | Dataset            | AG   | SMS Spam | IMDB | Tweet Emotion | Humor detection | Tweet Irony |
> |--------------------|------|----------|------|---------------|-----------------|-------------|
> | Subjectivity level | *    | ***      | **** | *****         | *****           | *****       |
> | real data          | 95.3 | 97.2     | 87.6 | 77.7          | 97.0            | 72.2        |
> | Original prompt    | 89.3 | 93.8     | 81.2 | 58.1          | 56              | 63.4        |
> | bad prompt         | 88.9 | 85.9     | 80.7 | 56.9          | 55.1            | 62.8        |
>
> we noticed that this ``bad prompt'' led to a slight decline in the effectiveness of the zero-shot synthetic data. Nevertheless, the overall trend in effectiveness of synthetic data remains consistent: synthetic data appears more effective for tasks with lower subjectivity, which is in align with the results from our original prompt.
>
> > It may be interesting to try (or to comment) on a setup with multiple generation chains where synthetic examples are used in the prompt rather than real example to generate more synthetic examples. If the overall quality degrades even further, that will be additional evidence that synthesizing examples can be considered a noisy channel.
>
> Thank you for your suggestion. We conducted an addition study by using the zero-shot synthetic data produced by the GPT-3.5 turbo model as the guiding example in a few-shot setting to generate data. As illustrated in Table 3 below,
>
> **Table 3:** Comparing the performance of Bert classification models trained on the zero-shot synthetic data and the few-shot synthetic data where the synthetic data is used as the guiding examples (``second-promot'') generated by the GPT-3.5 turbo in terms of Macro-F1 (\%).
> | Dataset            | AG   | SMS Spam | IMDB | Tweet Emotion | Humor | Tweet Irony |
> |--------------------|------|----------|------|---------------|-------|-------------|
> | Subjectivity level | *    | ***      | **** | *****         | ***** | *****       |
> | real data          | 95.3 | 97.2     | 87.6 | 77.7          | 97.0  | 72.2        |
> | Zero-shot          | 89.3 | 93.8     | 81.2 | 58.1          | 56.0  | 63.4        |
> | Second-prompt      | 87.1 | 86.9     | 81.1 | 55.9          | 53.8  | 61.9        |
>
> we found that for tasks with varying levels of subjectivity, using synthetic data as examples in the prompt for further synthetic data generation (referred to as ``second-prompt'') leads to a more  performance degradation compared to data generated in a single zero-shot round. This suggests that synthesized examples can act as a noisier channel in comparison to real examples.
>
> #### We will add the additional experimental results into the later paper or the appendix.

---

### Official Review · Reviewer_wx6Z · 2023-08-11

**Soundness:** 4

**Excitement:**

4: Strong: This paper deepens the understanding of some phenomenon or lowers the barriers to an existing research direction.

**Paper Topic And Main Contributions:**

This work investigates the potential of synthetically generated data in lieu of real data. They investigate effectiveness of synthetic data across a variety of classification tasks versus the "subjectivity" of the task. They use a variety of experiments and human evaluation to conclude that LLMs are good at generating data for "easier" datasets like news, reviews versus more complex datasets like humor.

**Questions For The Authors:**

While it is not a major issue, I would like to understand from authors their thoughts about using GPT-3.5 as the LLM here. GPT-3.5 has likely been trained on all of internet before 2021 atleast, and all the datasets used in this study are pre-2020. This means it has likely seen the data, including the test data of these datasets. I have three questions on this:

A. Is using GPT-3.5 really zero shot/few shot if it has seen these datasets already? Given how well LLMs memorize examples, I would be particularity be concerned about the few shot setting.

B. What's being thought of as subjectivity/complexity of the dataset, is the artifact of just the training data generator has seen?
 AG is news, IMDB are reviews -- one of the most common data types on the internet. Finance news and humor news are relatively niche and low in volume on internet (I would presume, I don't have data to back it up) compared to other datasets - perhaps needing special knowledge to understand. FPB in particular is a simple dataset but does need specialized knowledge to get sentiment; for example, does increasing interest rates is a negative event? I would doubt a MechTurk worker would had that perspective but someone with a knowledge of finance could have been easy to interpret that event easily. So what is being thought of subjectivity could be just about a) relative occurrence in LLMs training data; b) not as subjective for someone who knows the domain.

C. I don't think these undermine the study, but I would be interested in knowing what authors thing about this and should it be a part of limitations. Would you have seen different results with something like GPT-2 or BloombergGPT?

D. There is no limitation on amount of data one can generate using synthetic data generators. Studies (Aggarwal, 2023) have shown that using more synthetic data (2-10x real data) especially in low resource settings can outperform real data. Further, using weak supervision and simple data selection strategy (Chen, 2022) one can select complex examples which can outperform real data. Did you try generating more than just 1x real data to see if it can help in classification performance and perhaps other measures used in the study?


Aggarwal, K., Jin, H. and Ahmad, A., 2023. ECG-QALM: Entity-controlled synthetic text generation using contextual Q&A for NER.

Chen, M., Papangelis, A., Tao, C., Rosenbaum, A., Kim, S., Liu, Y., Yu, Z. and Hakkani-Tur, D., 2022. Weakly supervised data augmentation through prompting for dialogue understanding. arXiv preprint arXiv:2210.14169.

**Reasons To Accept:**

- Paper is very well written and easy to follow.
- This work tries to analyze the important problem of how good is synthetic data versus real data, which has been often overlooked. They have done an extensive study involving human evaluation to make their conclusions. This would be really valuable to researchers working in the synthetic data generation community.

**Reasons To Reject:**

- I do not see major flaws in this paper except using GPT-3.5 as the LLM. Please see questions for authors. Experiments are well motivated and authors have done a great job in adding requisite details in the paper with sufficient caveats, when required.
- The only major concern is that what authors are correlating with the complexity of the dataset might just be an artifact of familiarity in the training data of LLMs?

**Reproducibility:**

4: Could mostly reproduce the results, but there may be some variation because of sample variance or minor variations in their interpretation of the protocol or method.

**Reviewer Confidence:**

4: Quite sure. I tried to check the important points carefully. It's unlikely, though conceivable, that I missed something that should affect my ratings.

---

> ### Author Rebuttal · Authors · 2023-08-29
>
> Thanks for your insightful feedback. Below we address your main questions.
>
> > Is using GPT-3.5 really zero shot/few shot if it has seen these datasets already? Given how well LLMs memorize examples, I would be particularity be concerned about the few shot setting.
>
> Thank you for raising concerns that GPT-3.5 could potentially just memorize the training data examples. To address this concern, we conducted  an additional study on two post-2022 datasets: ChatGPT App reviews [1] (less subjective) and the 2023 Tweet Emotion dataset [2] (more subjective). We trained an Bert-based model on these two datasets. As shown in Table 1 below,
>
> **Table 1:** Comparing the performance of Bert classification models trained on the GPT-3.5 turbo-generated synthetic data under the
> zero-shot or few-shot settings, with those trained with the original real-world data, in terms of Macro-F1 (\%).
>
> | Dataset   | ChatGPT APP review | 2022 Tweet Emotion |
> |-----------|--------------------|--------------------|
> | real      | 79.4               | 68.9               |
> | zero-shot | 73.3               | 53.5               |
> | few-shot  | 76.5               | 58.8               |
>
>
> the results confirm our earlier findings: synthetic data is more effective in less subjective tasks like ChatGPT reviews, and adding real examples in the few-shot setting can improve zero-shot data efficacy.
>
> [1] ChatGPT App Reviews. https://www.kaggle.com/datasets/saloni1712/chatgpt-app-reviews
>
> [2] 2022 Tweet Emotion Reviews. https://www.kaggle.com/datasets/ankitkumar2635/sentiment-and-emotions-of-tweets
>
> > What's being thought of as subjectivity/complexity of the dataset, is the artifact of just the training data generator has seen? AG is news, IMDB are reviews -- one of the most common data types on the internet. Finance news and humor news are relatively niche and low in volume on internet (I would presume, I don't have data to back it up) compared to other datasets - perhaps needing special knowledge to understand. FPB in particular is a simple dataset but does need specialized knowledge to get sentiment; for example, does increasing interest rates is a negative event? I would doubt a MechTurk worker would had that perspective but someone with a knowledge of finance could have been easy to interpret that event easily. So what is being thought of subjectivity could be just about a) relative occurrence in LLMs training data; b) not as subjective for someone who knows the domain.
>
> Thanks for your constructive thought.
> 1. We do not think  ''subjectivitiy'' of tasks reflects simply the relative occurrence of different types of data in LLMs' training data.
> For instance, compared to its effectiveness in AG news or IMDB reviews, synthetic data generated by LLMs appears less effective for sentiment analysis of posts on online platforms like Tweet or Reddit (Tweet Emotion/Reddit Emotion), which are also common types of data on the Internet and likely be included in the LLMs' training data. Moreover, if the model performance degradation for those types of tasks with higher subjectivity are entirely caused by the limited occurrence of relevant data in LLMs' training data, one would expect that the performance degradation occurs on all task instances regardless of their (instance-level) subjectivity. However, this is different from what we actually observed in our second evaluation study, which suggests that the performance degradation on those types of tasks with high subjectivity mostly occurs on task instances with higher subjectivity.
>
> 2. We do not think the task subjectivity is just an artifact of that human subjects recruited in our study lack the domain expertise. In fact, even domain experts may find tasks that are deemed as subjective in our study truly ''subjective''. Take a real instance from the Finanical Phrase Bank--''Sales in Finland rose by 3.9\% and international growth was 0.7\% .''-- as the example. This event would be also very subjective for domain experts to determine the sentiment. For example, professionals overseeing the Finnish market might deem this positive as an affirmation of robust domestic strategies while global strategists could view it as the negative due to the modest 0.7\% international surge as a sign of market penetration challenges. Similarly, judging whether ``interest rates increase'' is positive or negative could also be subjective. For companies or investors grappling with substantial debt, an increase in interest rates could be perceived as negative, given that it would amplify their borrowing costs. On the other hand, those with cash reserves could see it as positive which is an opportunity to earn more interest.
>
> > I don't think these undermine the study, but I would be interested in knowing what authors thing about this and should it be a part of limitations. Would you have seen different results with something like GPT-2 or BloombergGPT?
>
> To examine whether our findings hold true for decoder-based models as well as models that are reasonably large,  we conducted the same evaluation studies using the GPT2-large (774M) and Llama2 (7B) models.
> Due to the time constraints, we conducted this evaluation on 6 randomly selected  datasets from the entire set of 10 datasets that we considered in our paper, and the selected 6 datasets covered different levels of subjectivity. As indicated in Table 2 below,
>
> **Table 2:** Comparing the performance of Bert classification models trained on synthetic data generated by various LLMs within a zero-shot setting using Macro-F1 (\%) as the metric.
> |                    |  AG  | IMDB |  SMS | Tweet Emotion | Humor Speech | Tweet Irony |
> |:------------------:|:----:|:----:|:----:|:-------------:|:------------:|:-----------:|
> | Subjectivity level |   *  |  *** | **** |     *****     |     *****    |    *****    |
> |      Real Data     | 95.3 | 87.6 | 97.2 |      77.7     |     97.0     |     72.2    |
> |    GPT2-large (774M)    | 86.5 | 80.9 | 86.4 |      52.2     |     51.5     |     60.8    |
> |    Llama 2 (7B)    | 88.9 | 82.4 | 88.5 |      59.1     |     57.2     |     63.1    |
> |    GPT-3.5 turbo   | 89.3 | 81.2 | 93.8 |      58.5     |      56.0      |     63.4    |
>
> we observed that models trained on the LLM-generated synthetic data only exhibits slight variations among different LLMs for each respective task.  The overall trend as seen in our paper remains consistent: the effectiveness of synthetic data tends to be higher for tasks with lower subjectivity. For the Bloomberg-GPT model or other domain-specific LLMs , due to the access limit, we can only conjecture that it can generate more effective synthetic data in the domains that require specific financial knowledge. However, this effectiveness might also be moderated by the subjectivity of the task or the task instance itself. For example, even experts might differ in their interpretation of a financial event such as "the interest increases".
>
> > There is no limitation on amount of data one can generate using synthetic data generators. Studies (Aggarwal, 2023) have shown that using more synthetic data (2-10x real data) especially in low resource settings can outperform real data. Further, using weak supervision and simple data selection strategy (Chen, 2022) one can select complex examples which can outperform real data. Did you try generating more than just 1x real data to see if it can help in classification performance and perhaps other measures used in the study?
>
> Thanks for pointing out these papers, we will add them in the reference. To see whether the more synthetic data can enhance classification performance, we conducted an ablation study based on GPT-3.5 turbo on several tasks with Bert as our base model for classification. Due to the time limit, we selected a subset of tasks from our main study and augmented the size of synthetic data to a maximum of three times that of the real data. As depicted in Table 3 below,
>
> **Table 3:** Comparing the performance of Bert classification models trained on varying size of the GPT-3.5 turbo-generated synthetic data under the
> zero-shot in terms of Macro-F1 (\%).
> | Ratio of synthetic data to real data size | 0.5  | 1    | 1.5  | 2    | 2.5  | 3    |
> |-------------------------------------------|------|------|------|------|------|------|
> | SMS Spam                                  | 92.9 | 93.6 | 93.4 | 93.2 | 93.1 | 91.8 |
> | AG                                        | 89.4 | 88.7 | 88.2 | 88.3 | 88.2 | 88.4 |
> | Tweet Emotion                             | 52.6 | 57.8 | 58.9 | 57.4 | 56.3 | 56.5 |
> | Financial                                 | 45.1 | 55.3 | 51.2 | 49.5 | 48.7 | 46.4 |
> | Sarcasm detection                         | 56.2 | 51.4 | 49.6 | 47.2 | 45.8 | 43.7 |
> | Relation                                  | 92.5 | 92.1 | 91.6 | 91.4 | 92.2 | 91.8 |
>
> our observations indicate that, unlike in low-resource settings, simply augmenting data by 0.5-3 times using unfiltered synthetic data does not consistently enhance performance across general tasks.
>
> #### We will add the additional experimental results into the later paper or the appendix.

---

### Official Review · Reviewer_M41S · 2023-08-12

**Soundness:** 4

**Excitement:**

3: Ambivalent: It has merits (e.g., it reports state-of-the-art results, the idea is nice), but there are key weaknesses (e.g., it describes incremental work), and it can significantly benefit from another round of revision. However, I won't object to accepting it if my co-reviewers champion it.

**Paper Topic And Main Contributions:**

The paper conducts a detailed analysis of the effect of subjectivity on classification tasks and its effect on zero-shot and few-shot-based synthetic data generated by LLMs.

**Questions For The Authors:**

1. The authors propose a “diversity prompt”. What measures have been taken to ensure the new data is diverse enough? Is it done using Remote Clique Score?

**Reasons To Accept:**

1. The paper is well written with a comprehensive explanation of how the subjectivity analysis was done.
2. Paper experimentally confirms the intuition of zero-shot synthetically generated data performs worse than few-shot synthetic data generation.
3. Conducted sound evaluation studies and presents acceptable explanations for the various findings as well as the few anomalies in the trends.

**Reasons To Reject:**

1. The paper does not conduct extensive experiments using decoder-based models as well as models that are reasonably large such as the 500M-750M parameters range. If these findings hold good for slightly larger models, it would be empirically conclusive proof of the findings.
2. Although the tasks are slightly different, the following papers mentioned empirically show the effectiveness of the LLM-generated dataset in a zero-shot setting which raises the question, if whether better querying/prompting of LLMs knowledge bases or synthetic data creation pipelines can, in fact, improve the performance of small models using zero-shot synthetic data.
* Generating Training Data with Language Models: Towards Zero-Shot Language Understanding (https://arxiv.org/pdf/2202.04538.pdf) - SuperGen
* SELF-GUIDED NOISE-FREE DATA GENERATION FOR EFFICIENT ZERO-SHOT LEARNING (https://arxiv.org/pdf/2205.12679.pdf) - SunGen

**Reproducibility:**

4: Could mostly reproduce the results, but there may be some variation because of sample variance or minor variations in their interpretation of the protocol or method.

**Reviewer Confidence:**

3: Pretty sure, but there's a chance I missed something. Although I have a good feel for this area in general, I did not carefully check the paper's details, e.g., the math, experimental design, or novelty.

---

> ### Author Rebuttal · Authors · 2023-08-29
>
> Thanks for your thoughtful review. Below we address your main questions.
>
> > The paper does not conduct extensive experiments using decoder-based models as well as models that are reasonably large such as the 500M-750M parameters range. If these findings hold good for slightly larger models, it would be empirically conclusive proof of the findings.
>
> To examine whether our findings hold true for decoder-based models as well as models that are reasonably large,  we conducted the same evaluation studies using the GPT2-large (774M) and Llama2 (7B) models.
> Due to the time constraints, we conducted this evaluation on 6 randomly selected  datasets from the entire set of 10 datasets that we considered in our paper, and the selected 6 datasets covered different levels of subjectivity. As indicated in Table1 below,
>
> **Table 1:** Comparing the performance of Bert classification models trained on synthetic data generated by various LLMs within a zero-shot setting using Macro-F1 (\%) as the metric.
> |                    |  AG  | IMDB |  SMS | Tweet Emotion | Humor Speech | Tweet Irony |
> |:------------------:|:----:|:----:|:----:|:-------------:|:------------:|:-----------:|
> | Subjectivity level |   *  |  *** | **** |     *****     |     *****    |    *****    |
> |      Real Data     | 95.3 | 87.6 | 97.2 |      77.7     |     97.0     |     72.2    |
> |    GPT2-large (774M)    | 86.5 | 80.9 | 86.4 |      52.2     |     51.5     |     60.8    |
> |    Llama 2 (7B)    | 88.9 | 82.4 | 88.5 |      59.1     |     57.2     |     63.1    |
> |    GPT-3.5 turbo   | 89.3 | 81.2 | 93.8 |      58.5     |      56.0      |     63.4    |
>
> we observed that models trained on the LLM-generated synthetic data only exhibits slight variations among different LLMs for each respective task.  The overall trend as seen in our paper remains consistent: the effectiveness of synthetic data tends to be higher for tasks with lower subjectivity.
>
> > Although the tasks are slightly different, the following papers mentioned empirically show the effectiveness of the LLM-generated dataset in a zero-shot setting which raises the question, if whether better querying/prompting of LLMs knowledge bases or synthetic data creation pipelines can, in fact, improve the performance of small models using zero-shot synthetic data.
>
> Thanks for pointing out these papers, we will add them in the reference and discuss them in the paper. To see how the improved data generation pipeline performs in tasks with different subjectivity levels, we conducted an ablation study by evaluating how SunGen [1] performs in tasks with varied subjectivity levels.
> As shown in Table2 below,
>
> **Table 2:** Comparing the performance of Bert classification models trained on synthetic data generated by the SunGen pipeline and our zero-shot pipeline using Macro-F1 (\%) as the metric.
> |                    |  AG  | IMDB |  SMS | Tweet Emotion | Humor Speech | Tweet Irony |
> |:------------------:|:----:|:----:|:----:|:-------------:|:------------:|:-----------:|
> | Subjectivity level |   *  |  *** | **** |     *****     |     *****    |    *****    |
> |      Real Data     | 95.3 | 87.6 | 97.2 |      77.7     |     97.0     |     72.2    |
> |       SunGen       | 91.7 | 84.7 | 94.5 |      61.8     |     59.9     |     64.6    |
> |  Zero-shot (Ours)  | 89.3 | 81.2 | 93.8 |      58.5     |      56      |     63.4    |
>
> while SunGen does offer an improvement compared to directly prompting LLMs for zero-shot synthetic data generation, the effectiveness of SunGen's synthetic data compared to real data is still influenced by the task's subjectivity level.
>
>
> > The authors propose a “diversity prompt”. What measures have been taken to ensure the new data is diverse enough? Is it done using Remote Clique Score?
>
> The diversity prompt serves as a constraint to instruct the LLM to produce more diverse data. We didn't compute the Remote Clique Score or similar metrics between newly generated data instances and the preceding $n$ instances in the generation process with a threshold to determine whether to accept the new generated data or prompt the LLM to regenerate it based on feedback.  The Remote Clique Score and Chamfer distance is used for the post-hoc analysis of diversity of data. We empirically found that without this diversity prompt, the Remote Clique Score  or Chamfer distance between new generated data instances and the previous $n$ instances often tend to be very low.
>
> #### We will add the additional experimental results into the later paper or the appendix.
>
> [1] SELF-GUIDED NOISE-FREE DATA GENERATION FOR EFFICIENT ZERO-SHOT LEARNING (https://arxiv.org/pdf/2205.12679.pdf) - SunGen

---

### Meta-Review · Area_Chair_fyA9 · 2023-09-18

**Recommendation:** 5

**Metareview:**

**quality, clarity, originality**

The paper is well written, as also stated by the reviewers. It is comprehensive and high quality. The authors set out the problem well and work to provide different views on the synthetic data generation and classification problems. The inclusion of a wide array of NLP classification tasks works in the favour of this paper. There were a number of reviewers who raised the challenge about the "subjective" vs "less subjective" datasets (the latter having less examples in the paper). The authors have addressed this by adding more experiments with more datasets that would be classified (by their measure) as "less subjective". This is welcome as it allowed better evaluation and understanding of the overall contribution of this paper. The final result might be something that many researchers had an intuition about, but this work provides an empirical and scientific study that confirms it and will form the basis (as I see) of how others also build on evaluating such approaches to data generation for their use cases.

** significance**

As stated earlier, the work gives very strong evidence about zero shot vs few shot performance for nlp classification data generation. It is a significant result that will be helpful in this LLM era. There was a concern about the LLMs covered for the experiments, but these have also been expanded and this adds to the pros of this paper.

**Notes to authors**

Will the generated data be made available for further evaluation by other groups?

---

### Decision · Program_Chairs · 2023-10-07

**Decision:**

Accept-Main

**Comment:**

**quality, clarity, originality**

The paper is well written, as also stated by the reviewers. It is comprehensive and high quality. The authors set out the problem well and work to provide different views on the synthetic data generation and classification problems. The inclusion of a wide array of NLP classification tasks works in the favour of this paper. There were a number of reviewers who raised the challenge about the "subjective" vs "less subjective" datasets (the latter having less examples in the paper). The authors have addressed this by adding more experiments with more datasets that would be classified (by their measure) as "less subjective". This is welcome as it allowed better evaluation and understanding of the overall contribution of this paper. The final result might be something that many researchers had an intuition about, but this work provides an empirical and scientific study that confirms it and will form the basis (as I see) of how others also build on evaluating such approaches to data generation for their use cases.

** significance**

As stated earlier, the work gives very strong evidence about zero shot vs few shot performance for nlp classification data generation. It is a significant result that will be helpful in this LLM era. There was a concern about the LLMs covered for the experiments, but these have also been expanded and this adds to the pros of this paper.

**Notes to authors**

Will the generated data be made available for further evaluation by other groups?